



# Spatially aggregated climate indicators over Sweden (1860–2020), part 2: Precipitation

Christophe Sturm[1]

[1]Hydro-climatology research group, Swedish Meteorology and Hydrology Institute

**Correspondence:** Christophe Sturm (christophe.sturm@smhi.se)

**Abstract.**

The Swedish Meteorology and Hydrology Institute (SMHI) provides a national aggregated climate indicator from 1860 to present. We present a new method to compute the national climate indicator based on Empirical Orthogonal Functions (EOF). EOF are computed during the 1961 – 2018 calibration period, and later applied to the full experiment period 1860–2020. This
study focuses the climate indicator for precipitation; it follows the same methodology as for the national climate indicator for temperature, described in the companion article (Sturm, 2024a).

The new method delivers results in good overall agreement with the reference method (i.e. arithmetic mean from selected stations in the reference network). Discrepancies are found prior to 1900, primarily related to the reduced number of active stations: the robustness of the indicator estimation is assessed by an ensemble computation with added random noise, which
confirms that the ensemble spread increases significantly prior to 1880.

The present study establishes that the 10-year running averaged precipitation indicator rose from $-8.37$ mm.month$^{-1}$ in 1903 to $4.08$ mm.month$^{-1}$ in 2010 (with respect to the mean value of $54.18$ mm.month$^{-1}$ for the 1961–2018 calibration period), i.e. a 27% increase over a century. Winter (DJF) precipitation rose by $+20$ mm.month$^{-1}$ between 1890–2010, summer precipitation by $+25$ mm.month$^{-1}$.

The leading EOF patterns illustrate the spatial modes of variability for climate variability. For precipitation, the first EOF pattern displays more pronounced regional features (maximum over the West coast), which is completed by a north-south seesaw pattern for the second EOF. We illustrate that EOF patterns calculated from observation data reproduce the major features of EOF calculated from GridClim, a gridded dataset over Sweden, for annual and seasonal averages. The leading EOF patterns vary significantly for seasonal averages (DJF, MAM, JJA, SON) for precipitation.

Finally, future developments of the EOF-method are discussed for calculating regional aggregated climate indicators, their relationship to synoptic circulation patterns and the benefits of homogenisation of observation series.

The EOF-based method to compute a spatially aggregated indicator for temperature is presented in a companion article (Sturm, 2024a), which includes a detailed description of the datasets and methods used in this study

. The code and data for this study is available on Zenodo (Sturm, 2024b).



## 1 Introduction

The latest IPCC – Physical science basis report (Masson-Delmotte et al., 2021; Gulev et al., 2021)) emphasises the importance of instrumental climate records: they provide an unequivocal proof of the ongoing climate change. At a national level, a spatially aggregated climate indicator illustrates the climate variability: it is a relevant tool for researchers, decision-makers and the general public.

Establishing a national climate indicator (hereafter referred to as **CI**) requires a numerical methods that aggregate time-series from individual observation stations over larger area into a single time-series. The goal of the present method is to compute a bias-free estimator, despite the fact that data availability (i.e. station activity) varies over time. In other words, the challenge is to define a uniform method, whose results for the early part of part of the record (with only few active observation stations, not necessarily spread uniformly over the territory) will be consistent with later results (with numerous, evenly spaced stations).

Defining a climate indicator for precipitation is more challenging than for temperature, as presented in Sturm (2024a). First, the de-correlation distance (i.e. the distance between to stations where observations are no longer significantly correlated) is significantly smaller for precipitation than temperature. This requires thus a higher station density to account for the spatial heterogeneities. Second, the geographical patterns for the departure from climatology of annual and seasonal precipitation means display larger regional differences than for temperature. Therefore, the national climate indicator for precipitation is

likely to be more sensitive to the station locations (i.e. the evolution of the station network) than for temperature.

    The Swedish Meteorology and Hydrology Institute (SMHI) has collected and compiled quality-ensured observations of precipitation across Sweden since the eighteenth century: in the current study, we present a total of 2115 time-series for precipitation over the 1860–2020 period from observation stations across Sweden. The new method, described extensively in Sturm (2024a), uses a gridded climate dataset (GRIDCLIM (Andersson et al., 2021)), available over 1961–2018, to replace

missing observation values . The (now complete) calibration dataset of observations are analysed with Empirical Orthogonal Functions (EOF).

    The analysis of the EOF patterns for annual and seasonal means reveal new insights into the characteristics of precipitation over Sweden. Leading EOF patterns computed from the full GRIDCLIM gridded dataset illustrate the behaviour of poorly sampled areas, e.g. in remote, high-elevation areas. In order words, this study evaluates the representativeness of the MORA

observation network for the entire Swedish territory.

    In order to test the robustness of the new climate indicator, we blend random noise and random sub-sampling from the original station observations: this ensemble computation allows to define the 25% and 75% percentiles (in other words, the range between which 50% of all ensemble computations are found).

    Finally, we discuss the discrepancies between the new EOF-based method and the original SMHI method, using an arithmetic

mean of observations from the reference station network.

## 2 Data and methods

We refer to the companion article (Sturm, 2024a) for detailed description of the datasets used in this study





. The most important aspects are listed hereafter.

Two datasets are used in this study: SMHI's database for station observations over 1860–2020 called *SMHI-MORA*, and the gridded dataset *SMHI-GridClim* (Andersson et al., 2021) produced as a combination of *SMHI-MORA* observations with *UERRA* regional reanalysis over the 1961–2018 period. The datasets *SMHI-MORA* and *SMHI-GridClim* will hereafter be referred to as MORA and GRIDCLIM.

The method currently used to compute the national climate indicator for precipitation is based on an arithmetic average of available observations: we recall the reference method, along with its underlying equations. We then introduce the new method based on empirical orthogonal functions (EOF), computed over the 1961–2018 period for which *UERRA* reanalysis is available. Leading EOF patterns established over the 1961–2018 period are used to reconstruct the climate indicator over the MORA period 1860–2020.

We finally describe a method to evaluate the robustness of various climate indicator estimators: the span of estimator realisations is computing by adding normally-distributed noise and restricting the computation to a subset of MORA stations.

## 2.1 Observational data sets

### 2.1.1 MORA observation station network

The MORA [1] database centralises meteorological and hydrological observations in Sweden. Each station in MORA is identified with a unique numeric ID, station name, geographic coordinates (longitude, latitude, altitude) as meta-data. If a station is relocated (e.g. if increasing nearby urbanisation leads to the station no longer), a new station ID is created; after a few years overlap, the old station is usually closed. The MORA thus contains many missing values, as illustrated in Fig. (1). Table (1) summarises the maximum number of simultaneously active stations –which occurs in the early 1960's– amounts to 46.5% for precipitation of the total number of stations in the network.

The lower plot in Fig. (1) indicate the median latitude (as thick red line) and [25%; 75%] percentiles (in thin dashed red lines) of active stations over time. The median latitude is used in the present study as a proxy for the progressive extension of the observation network, in particular the increased station density in Sweden's northern regions over 1860–2020. It is worth noticing that the relative decrease of active observations in MORA since 1960 proportionally favoured observation stations in northern parts of Sweden.

Historical observations are continuously added to the MORA database, with ongoing digitalisation of historic observation reports (from printed records). Newly rescued observations are quality-controlled, but currently not (yet) homogenised: hence possible biases related to changes in measure instruments or modifications in the station's environment are not corrected.

The present study considers annually resolved observations, representing the annual average (hereafter referred to as *ANN*), or seasonal averages [2] (winter – *DJF*, spring – *MAM*, summer – *JJA* and autumn – *SON*).

---

[1] The MORA acronym stands for "Meteorologiska observationer för realtid och arkiv", meaning "Meteorological observations for real-time and archive".

[2] In season's acronyms, the capital letter represents the first letter of the month (e.g. DJF: December–January–February).



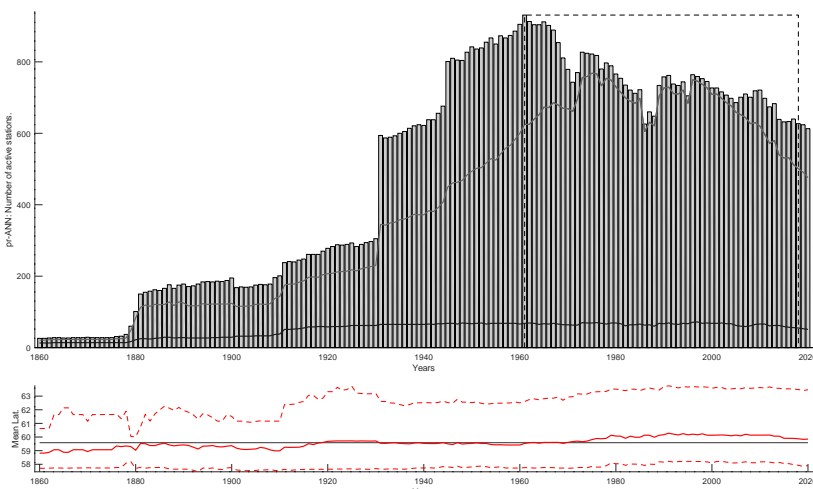

**Figure 1.** Upper plot: Number of active temperature stations in MORA over time (as bars). The dark grey line represents the number of active in the original reference station network; they light grey line represents the number of stations for the calibration network (i.e. individual stations being active at least 15 years during the calibration period 1961–2018, as highlighted by the dashed box). Lower plot: Median latitude for active stations in the calibration dataset over time (incl. the [25%–75%] bounds). The median latitude is used as a proxy for the distribution of the observation network.

Observations in the MORA database extend to the mid-18[th] century for selected stations, e.g. Uppsala since 1722 (Bergström and Moberg, 2002), Stockholm since 1756 (Moberg and Bergström, 1997) or Lund since 1780 (Bärring et al., 1999). In the present study, we select 1860 as initial year for climate indicator, which corresponds to establishment of the first meteorological observation network by the Swedish Academy of Science (Svenska Vetenskapsakademin) in 1858–1860. For the present study, we retrieved precipitation observations for all available stations over the 1860–2020 period, focusing on *annually-resolved* climate indicators: annual (ANN) and seasonal (winter DJF: December–February, sprint MAM: March–May, summer JJA: July–August, autumn SON: September–November) averages are calculated from monthly values, retrieved from the MORA database, as arithmetic average, provided that no month was missing. The number of stations for precipitation measurements are indicated in Table (1), while the evolution of active stations is shown in Fig. (1). In Table (1), the fist column indicates the number of stations in the MORA precipitation dataset. The rows *Initial* and *Present-day* show the number active stations respectively at the start and the end of observation series. Additional rows GridClim[start] and GridClim[end] indicate the number of active stations during the 1961–2018 calibration period. The row *Maximum* indicates the year for which the number of active stations was maximal, along with the corresponding number of active stations.



| Dataset | Period | $Y_{max}^{prec}$ | # Sta. | Perc. |
|---------|--------|------------|--------|-------|
| Complete | Total | | 2115 | 100% |
| | Initial | 1860 | 26 | 1.2% |
| | Maximum | 1962 | 984 | 46.5% |
| | Present-day | 2020 | 660 | 31.2% |
| Calibration | Total | | 1098 | 100% |
| | Initial | 1860 | 21 | 1.9% |
| | GridClim$^{start}$ | 1961 | 632 | 57.6% |
| | Maximum | 1975 | 808 | 73.6% |
| | GridClim$^{end}$ | 2018 | 521 | 47.4% |
| Reference | Total | | 87 | 100% |
| | Initial | 1860 | 13 | 14.9% |
| | Maximum | 1996 | 75 | 86.2% |
| | Present-day | 2020 | 55 | 63.22% |

**Table 1.** Number of stations found in the MORA database. The upper third represents the **Complete** MORA dataset, the middle third the **Calibration** sub-set, i.e. stations with at least 15 years coverage during 1961–2018, and the bottom third the **Reference** selected stations for the original SMHI methodology. For each subset, **"Total"** indicates the number of stations active at least once during the 1860–2020 period; **"Initial"** shows the number of stations active in 1860; **"Maximum"** the maximum of simultaneously active stations; **"Present-day"** the number of stations currently active. The complete time-evolution of active stations can be found in figures Fig. (1).

### 2.1.2 GRIDCLIM gridded dataset

The GRIDCLIM project (Andersson et al., 2021), conducted at SMHI, combines the regional European reanalysis UERRA (Schimanke and Service, 2019; Schimanke et al., 2019) with station observations to produce a uniform, bias-corrected gridded climate dataset. Additional details on GRIDCLIM and underlying methodology are described in the companion article (Sturm,
2024a).

The GRIDCLIM dataset (Andersson et al., 2021) covers the period 1961–2018, in accordance with the availability of the UERRA reanalysis (Schimanke and Service, 2019; Schimanke et al., 2019). This is henceforth defined as the calibration period (1961–2018). As for the MORA dataset described above, precipitation from the GRIDCLIMdataset is averaged to annually-resolved annual (ANN) and seasonal (DJF, MAM, JJA, DJF) time-series.
In the following sections, GRIDCLIM dataset both as a gridded dataset covering Sweden, as well as emulations of the reference and larger MORA station networks. GridClim-all represents all $69\,842$ grid-cells of the GRIDCLIM dataset covering the Swedish territory (in its native EPSG:9001 Lambert Conic Conformal projection with $63°$ standard parallel, 2.5km x 2.5km resolution), is used to compute the EOF and SVD output. GridClim-all, i.e. **CI** computed as arithmetic mean of precipitation



over Sweden, is compared to SMHI-ref and other *CI* estimates. GridClim-sub is a subset of GridClim-all, whose grid-cells

correspond to MORA calibration station network. GridClim-sub is used for the gap-filling of MORA calibration dataset (1098 for precipitation) prior to the EOF analysis. GridClim-ref is a subset of GridClim-all, with time-series taken from grid-cells corresponding to the MORA reference network SMHI-ref (87 for precipitation).

## 2.2 Computing a national climate indicator

The theoretical assumptions and demonstration underlying the EOF method for the reconstruction of a spatially aggregated

climate indicator (*CI*) are described in detail in a companion article (Sturm, 2024a). We summarise hereafter the most important conclusions.

### 2.2.1 Emulating the operational SMHI method

SMHI provides climate indicators, including for precipitation, as part of its services for the community (Engström, 2022, 2023). The original method to compute the climate indicator (*CI*) is the simplest way to synthesise multiple stations records as a

single time-series: it uses an arithmetic average, as presented in Eq. (1), where $x_{i_{sta}}(t)$ represents an individual (incomplete) observation time-series and $n_{sta}(t)$ is a function counting active stations over time. The method is described in further detail in the online documentation (Engström, 2022, 2023). The reference observation network across the country consists of 87 for precipitation, cf. Table (1). The location of reference stations is indicated as black crosses in Fig. (2). In the results section, this dataset is referred to as SMHIRef.

$$
\begin{aligned}
CI_{ori}(t) &= \frac{\sum_{i_{sta}=1}^{n_{sta}} x_{i_{sta}}(t)}{n_{sta}(t)} \\
\Leftrightarrow CI_{ori}(j) &= \frac{\sum_{i=1}^{n} \mathbf{X}(i,j)}{\sum_{i=1}^{n} \left( \mathbf{X}(i,j) \neq \text{NaN} \right)}
\end{aligned}
\tag{1}
$$


In the present study, we emulate the original (arithmetic) method by extracting time-series from GRIDCLIM grid-cells in which the 87 reference stations are located: this estimator for the calibration period 1961–2018 is referred to as GridClim-ref in the result sections. Similarly, GridClim-sub consists of 1098 time-series (cf. Table (1)) extracted ad grid-points corresponding to the MORA calibration network. The estimator for mean precipitation over the entire Swedish territory (for the 1961–2018

period) is based on the arithmetic mean of all 69,852 grid-cells, referred to as GridClim-all.



### 2.2.2 EOF decomposition of 2-dimensional $[geographical, time]$ dataset

Prior to performing the EOF analysis over the calibration period 1961–2018, the observation dataset $\mathsf{X}_{MORA}$ is gap-filled using a linear regression of available observations versus the collocated, complete $\mathsf{X}_{GridClim}$ time-series:

$$\forall i \in [[1, n_{stations}]], \forall j \in [[1, n_{years}]],$$

$$\begin{cases} X_{MORA}(i,j) \neq \text{NaN} \Rightarrow & \mathcal{A}(i,j) = 1 \\ X_{MORA}(i,j) = \text{NaN} \Rightarrow & \mathcal{A}(i,j) = 0 \end{cases}$$

Linear regression MORA vs GRIDCLIM:

$$\forall i \in [[1, n_{stations}]], \forall j \in (\mathcal{A}(i,:) = 1),$$

$$\alpha^i \cdot \mathbf{X}_{GridClim}(i,:) + \beta^i + \varepsilon = \mathbf{X}_{MORA}(i,:) \tag{2}$$

Substitution of MORA missing values:

$$\forall i \in [[1, n_{stations}]], \forall j \in (\mathcal{A}(i,:) = 0),$$

$$\mathbf{X}_{MORA}(i,:) = \alpha^i \cdot \mathbf{X}_{GridClim}(i,:) + \beta^i$$

Centering MORA around 1961–2018 mean:

$$\mathbf{X_c}(i,:) = \mathbf{X}(i,:) - \overline{\mathbf{X}_{1961-2018}(i,:)}$$

Hence, the calibration sub-set of MORA stations is now complete over the 1961–2018 calibration period, with a zero mean over the 1961–2018 calibration period. Centering the dataset prior to applying the EOF analysis provides the advantage of representing the spatial and temporal variability of the studied dataset (i.e. respective spatial patterns and time loadings) as departures from zero.

The Empirical Orthogonal Functions (EOF) method, equivalent to Principal Component Analysis (PCA), aims at decom-

posing the spatio-temporal variability in the dataset $\mathbf{X}_c$ as a series of spatial patterns (hereafter referred to as spatial EOF patterns) $\mathbf{EOF}$, associated to its time expansion coefficients $\mathbf{A}$ (Björnsson and Venegas, 1997; Benestad et al., 2023; Navarra and Simoncini, 2010; Thomson and Emery, 2014; Wilks, 2011; Zhang and Moore, 2015).

$$\text{Covariance matrix: } \mathbf{R} = \mathbf{X}_c^T \cdot \mathbf{X}_c$$

$$\text{Eigenvalue problem: } \mathbf{EOF} \cdot \mathbf{R} = \mathbf{EOF} \cdot \mathbf{\Lambda}$$

$$\Leftrightarrow \mathbf{R} = \mathbf{EOF} \cdot \mathbf{\Lambda} \cdot \mathbf{EOF}^{-1}$$

$$\text{Time expansion coefficient: } \mathbf{A} = \mathbf{X}_c \cdot \mathbf{EOF} \tag{3}$$

As a result, the original matrix $\mathbf{X}_c$ can be identically reconstructed based on its decomposition in spatial patterns ($\mathbf{EOF}$)

and associated time expansion coefficients ($\mathbf{A}$). Assuming that the original dataset $\mathbf{X}_c$ has $n$ stations/grid-points, and $m$ time steps (as columns), the number of unique modes in the present study is set to $p = \min(n, m)$, i.e. in practice the number of





time steps.

$$\mathbf{X}_c = \mathbf{A} \cdot \mathbf{EOF}^T$$
$$\Leftrightarrow \mathbf{X}_c = \sum_{i=1}^{p} \overrightarrow{\mathbf{a}^i} \cdot \overrightarrow{\mathbf{eof}^{Ti}} \tag{4}$$

The EOF decomposition was applied over the 1961–2018 calibration period for the GRIDCLIM dataset, restricted to the
Swedish territory in Fig. (2). Analogue results of the EOF decomposition on the MORA calibration dataset for the 1961–2018
period is shown in Fig. (3).

### 2.2.3 Estimating the climate indicator and related uncertainties

The formulation in Eq. (4) can be extended beyond the 1961-2018 calibration period, despite missing values found in the
longer 1860–2020 record. Hereafter, $\mathbf{X}^\star$ represents a matrix $\mathbf{X}$ with missing values, and $\widehat{\mathbf{A}}$ the estimator of its time expansion
coefficients. In other words, $\widehat{\mathbf{A}}$ is an approximation of the time expansion coefficient matrix $\mathbf{A}$ for the entire study period
1860–2020 and full rank of $\Lambda$ (i.e. $p$, in our case the number of years in the calibration period, cf. Eq. (3)). Additional details
on the methodology to compute the national climate indicator are described in the companion article (Sturm, 2024a).

Eq. (5) expresses the new method for estimating the gap-filled observation dataset, based on the EOF decomposition of the
MORA calibration subset $\mathbf{X}_c^{MORA,Cal}$.

Time expansion coefficients:
$$\mathbf{A} \simeq \mathbf{X}_c^\star \cdot \mathbf{EOF}$$
$$\Leftrightarrow \widehat{\mathbf{A}} = \sum_{i=1}^{p} \mathbf{X_c^{\star i}} \cdot \mathbf{eof}^i$$
$$\Rightarrow \forall i \in [[1,p]]$$
$$\widehat{\mathbf{a}^i(t)} = \sum_{x_c^i(t) \neq \text{NaN}} x_c^i(t) \cdot \mathbf{eof}^i(t)$$

Dataset reconstruction:
$$\widehat{\mathbf{X}}_c^{EOF} = \widehat{\mathbf{A}} \cdot \mathbf{EOF}^{-1} \tag{5}$$


Since most of the variance is comprised in the leading modes (given that, by construction, eigenvalues in $\Lambda$ are listed in
decreasing order), we chose to restrict the reconstruction of $\widehat{\mathbf{A}}$ to the 10 leading modes (cf. Table Table (2)). Furthermore,
leading modes ($\mathbf{eof}^{MORA}(\#1-3)$ as shown in Fig. (2)) display spatial patterns consistent with climate phenomenon, while
modes with lower eigenvalues mostly display numerical "noise" (i.e. patterns without obvious physical significance). Hence
restricting to the 10 leading modes (i.e. $p=10$ in Eq. (5)) reduces the risk of spurious, physically inconsistent artefacts in
$\widehat{\mathbf{A}}^{MORA}$.



Analogously to Eq. (1) defining $CI_{ori}$, the EOF-based climate indicator $CI_{EOF}$ can be expressed as:

$$\forall j \in [[1, 2020 - 1860 + 1]], CI_{EOF}(j) = \frac{\sum\limits_{i=1}^{n} \widehat{\mathbf{X}}_c^{EOF}(i,j)}{\sum\limits_{i=1}^{n} (\mathbf{X}_c(i,j) \neq \mathrm{NaN})} \qquad (6)$$

Compared to the original method in Eq. (1), the new methods has the advantage that available observations contribute to
the estimated indicator according to their "weight" in the 10 leading modes. Hence the EOF-based method (Eq. (6)) can be
considered as a weighted, rather than arithmetic average of station data (Eq. (1)). Thus, it has the potential of being less sensitive
to changes in station coverage, as shown in Fig. (1).

In order to test the robustness of the climate indicator estimator, we perform an ensemble computation to assess the effect of
sub-sampling and added random noise.

$$\forall k \in [[1, n_{ens}]], CI_{ens}^{k}(j) = \frac{\sum\limits_{i \in (\text{k-th random subset})} \left( \widehat{\mathbf{X}}_c(i,j) + \varepsilon^k(i,j) \right)}{\sum\limits_{i \in (\text{k-th random subset})} (\mathbf{X}_c(i,j) \neq \mathrm{NaN})} \qquad (7)$$

Assuming that the daily observation errors are normally distributed around 0 (with a –conservative– estimate of measurement
uncertainty of $\varepsilon_{\mathrm{prec}} = \pm 10 \, mm.day^{-1}$ for precipitation), their corresponding error for monthly means are reduced by a factor
$\sqrt{30} \cdot \sqrt{n_{\mathrm{month}}}$, i.e. the average amount of daily measurements in a monthly mean. $n_{\mathrm{month}}$ represents the number of monthly
records in the annually-resolved average: for annual means, $n_{\mathrm{month}} = 12$, for seasonal means, $n_{\mathrm{month}} = 3$. Hence the random
noise function $\varepsilon$ is normally distributed, with a standard deviation of $\sigma_{ANN}^{\mathrm{precipitation}} = \frac{10}{\sqrt{30} \cdot \sqrt{n_{\mathrm{month}}}} = 0.53$ mm.month$^{-1}$ for annual
mean precipitation and $\sigma_{seas}^{\mathrm{precipitation}} = 1.05$ mm.month$^{-1}$ for seasonal means.

The second aspect to be assessed is the impact of sample size on the **CI** climate indicator estimate. A random subset of
stations within the calibration network, including three times as many stations as in the reference network were used to compute
the **CI**, with a new realisation of the random noise function $\varepsilon$. In other words, Table (1) indicates that the reference network
contains $n_{Ref} = 87$ for precipitation; the **CI** for each ensemble member is thus computed from $n = 3 \times n_{Ref}$ randomly chosen
stations within the calibration network.

The procedure above is repeated 100 times; the 25% and 75% percentiles are computed and presented as thin lines on Fig.
(8) for the centered **CI**, and Fig. (6) for the departure from the original SMHI-ref ($\Delta CI$). The same procedure was applied
to evaluate the 25% and 75% percentiles for GridClim-ref, albeit using 100 realisations with half (i.e. $n = \frac{n_{Ref}}{2}$) of available
stations in the reference network.

## 3 Results

Following figures represent the leading EOF modes of annual precipitation from GRIDCLIM (Fig. (2)); annual precipitation
from MORA (Fig. (3). seasonal precipitation from GRIDCLIM Fig. (4).

Similarly, the SVD decomposition was performed for precipitation for the GRIDCLIM and MORA datasets. The EOF and
SVD patterns are virtually identical. Hence the corresponding figures are shown in the supplementary material.



| | Dataset | Method | $\lambda(\#1)$ | $\lambda(\#2)$ | $\lambda(\#3)$ | Sum |
|---|---|---|---|---|---|---|
| P | GRIDCLIM | EOF | 45.1% | 17.9% | 8.7% | 71.8% |
| P | MORA | EOF | 43.2% | 12.4% | 7.9% | 63.5% |

**Table 2.** Portion of explained variance $\lambda$ for the leading three modes with the EOF analysis of GRIDCLIM and MORA over the 1961–2018 calibration period. The sum of the variance explained by the first 3 modes is indicated in the last column.

Table (2) indicates how much of the total variance is expressed in the leading three modes (as obtained from matrix $\Lambda$ in Eq. (3)).

### 3.1 EOF patterns for precipitation

By construction, plain (i.e. unrotated) spatial EOF patterns (without prior detrending) are expected to display a uni-modal distribution in the first mode, a bi-modal distribution in the second, and a tri-modal in the third. This behaviour is clearly apparent in Fig. (2).

Fig. (3) represent the EOF patterns for the MORA dataset. The polygons represent the Delaunay triangulation of the MORA stations in the calibration subset; it is worth noticing that, despite the area of the polygons vary (with a tendency to increase in northern Sweden), each time-series has the same weight in the EOF method.

The uni-modal pattern for the first mode highlights regions with high precipitation: Sweden's West coast (around Göteborg) and the North-Eastern Baltic coast (around Umeå). The bi-modal distribution of the second mode highlights the contrast between Scandic mountain range at the boarder with Norway and the rest of low-lands. The leading EOF modes for GRIDCLIM precipitation are shown in Fig. (2), and the corresponding results for MORA precipitation in Fig. (3).

#### 3.1.1 Annual EOF patterns for precipitation

The spatially less homogeneous character of precipitation causes the leading modes to retain less of the total variance than for temperature. Table (2) shows that GRIDCLIM (Fig. (2)) retains a comparable portion of the total variance ($\lambda_{\mathrm{GridClim}}^{EOF}(\#1) = 45.1\%$, $\sum_{i=1}^{3} \lambda_{\mathrm{GridClim}}^{EOF}(i) = 71.8\%$) as MORA's (Fig. (3)) leading modes ($\lambda_{MORA}^{EOF}(\#1) = 43.2\%$, $\sum_{i=1}^{3} \lambda_{MORA}^{EOF}(i) = 63.5\%$). It is worth noticing that the leading 10 modes, on which the reconstruction is based, respectively explain $\sum_{i=1}^{10} \lambda_{\mathrm{GridClim}}^{EOF}(i) = 88.4\%$ and $\sum_{i=1}^{10} \lambda_{MORA}^{EOF}(i) = 80.4\%$: hence, even for precipitation, the portion of variance is deemed sufficient to reconstruct $\widehat{\mathbf{X_c}}$ for the purpose of calculating the climate indicator $\boldsymbol{CI}$.

The first mode for annual precipitation $\mathbf{eof}_{\mathrm{GridClim}}(\#1)$ is characterised by two areas of increased precipitation (when the time-expansion coefficients are positive): Sweden's West coast (around Göteborg) and the North-Eastern Baltic coast (around Umeå), extending to the mountain region around Kebnekaise. By contrast, South-Eastern Sweden (around Kalmar) and Central-Eastern region (around Uppsala) tend to experience less precipitation increase than the country's average.





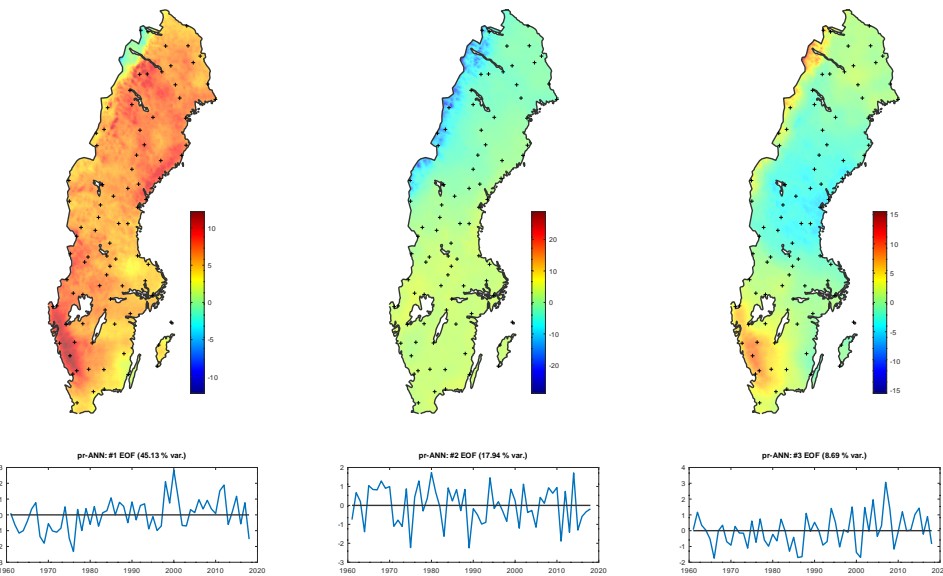

**Figure 2.** Leading three EOF patterns for GRIDCLIM, all grid-points covering Sweden, ($\mathbf{eof}^{\text{GridClim}}(\#1-3)$), with their associated time expansion vectors ($\mathbf{a}^{\text{GridClim}}_{EOF}(\#1-3)$) for *precipitation*. The unit of the $\mathbf{a}(\#1-3)\cdot\mathbf{eof}^{T}(\#1-3)$ product is $mm/month$.

It is interesting to notice that a small area in uttermost North-Western Sweden (from Abisko to Kvikkjokk) displays EOF values around 0. It indicates that the precipitation variability in this region, on the wind-side of the Sarek and Stora Sjöfallet massifs, is decoupled from the dominant mode over mainland Sweden. The precipitation variability in this region, and the Fennoscandian mountain range in general, is very pronounced in the following modes $\mathbf{eof}^{\text{GridClim}}_{\text{ANN}}(\#2)$ (for the entire mountain range) and $\mathbf{eof}^{\text{GridClim}}_{\text{ANN}}(\#3)$ (for the Sarek region in particular).

Fig. (3) shows the three leading EOF for precipitation in MORA (using the station network used for the calibration dataset EOF-rec). The overall features observed in GRIDCLIM's leading EOF modes are well reproduced, with comparable respective portion of explained variance ($\sum\lambda(\#1-3)=63.5\%$, cf. Table (2)). In particular, the first EOF pattern $\mathbf{eof}^{MORA}(\#1)$ illustrates a more intense precipitation increase over South-Western Sweden (Göteborg region) and North-East (Umeå). However, the second EOF pattern $\mathbf{eof}^{MORA}(\#2)$, $\lambda=13.11\%$ fails to capture the precipitation variability over the Fennoscandian mountain range bordering Norway, as stands out in $\mathbf{eof}^{\text{GridClim}}(\#2)$. The particular behaviour of the Sarek region (and high-altitude areas in general) only appear in $\mathbf{eof}^{MORA}(\#3)$, $\lambda=8.24\%$; in other words, the peculiarity of precipitation variability of mountain region is partially captured with EOF on the MORA dataset, but with less distinctive character than using the full GRIDCLIM dataset.





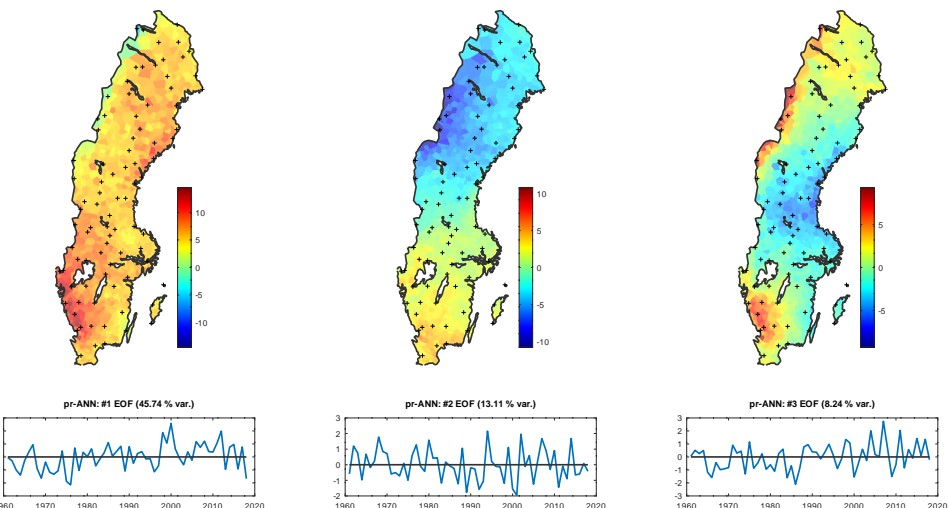

**Figure 3.** Leading three EOF patterns for MORA (Calibration network) *precipitation* over Sweden ($\mathbf{eof}^{MORA}(\#1-3)$), with their associated time expansion vectors ($\mathbf{a}_{EOF}^{MORA}(\#1-3)$) for precipitation.

This result is not surprising: as the location of stations in the reference observation network (marked as black crosses in Fig. (2)) indicate, high altitude regions are under-sampled compared to less remote, higher population density regions.

### 3.1.2 Seasonal EOF patterns for precipitation

As discussed in the previous section, the dominant precipitation EOF modes calculated from the GRIDCLIM dataset ($\mathbf{eof}^{\text{GridClim}}(\#1-3)$) are similar to those computed from the calibration network in MORA ($\mathbf{eof}^{MORA}(\#1-3)$), with the exception of high altitude regions in the Fennoscandian mountain range bordering Norway. The same holds for seasonal results; hence we will focus the discussion on the GRIDCLIM precipitation patterns ($\mathbf{eof}^{\text{GridClim}}(\#1-3)$).

The seasonal EOF decomposition of precipitation in the GRIDCLIM dataset is shown in Fig. (4), with DJF (winter, upper left panel), JJA (summer, upper right panel), MAM (spring, lower left panel) and SON (autumn, lower right panel).

The seasonal variability of precipitation displays large variations compared to the annnual EOF patterns (Fig. (2)). The winter (DJF) variability exacerbates regions with higher precipitation increase (Sweden's West coast around Göteborg and North-Eastern coast around Umeå): if the time expansion coefficient is positive, these regions receive up to 3-times as much precipitation as most of mainland Sweden: the colour-bar indicates that the EOF value (red) exceeds $30$ mm.month$^{-1}$, compared to the median of $5$ mm.month$^{-1}$, given that $\lambda_{DJF}^{\text{GridClim}}(\#1) = 50\%$.





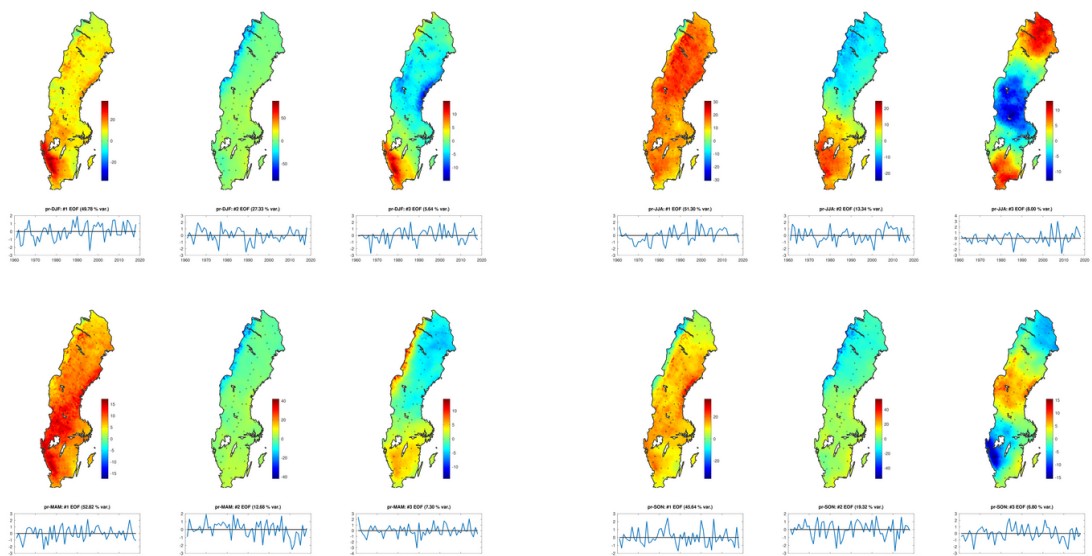

**Figure 4.** Leading three EOF seasonal patterns for GRIDCLIM *precipitation* over Sweden ($\mathbf{eof}(\#1-3)$, upper row), with their associated time expansion vectors ($\mathbf{a}(\#1-3)$, lower row) for temperature. The unit for the product $\mathbf{a}(\#1-3) \cdot \mathbf{eof}^T(\#1-3)$ is in mm.month$^{-1}$. Black crosses indicate the location of MORA stations in the reference network.

On the contrary, the leading EOF for summer $\mathbf{eof_{JJA}}(\#1)$ and spring $\mathbf{eof_{MAM}}(\#1)$ show a much less pronounced geographical pattern: the South-East coast (around Kalmar, including the islands of Öland and Gotland) and Eastern midlands

(around Uppsala) appear as slightly drier than the average.

Precipitation variability for autumn $\mathbf{eof_{SON}}(\#1)$ distinctively illustrates the decoupling of precipitation regimes over the Fennoscandian mountain range bordering Norway (primarily captured in $\mathbf{eof_{JJA}}(\#2)$), while increased precipitation is concentrated on the East coast (from Gävle to Umeå) and, to some extent, the South-West coast. This pattern is similar to

### 3.2 Reconstructing a national climate indicator

Several methods are evaluated to estimate the climate indicator (***CI***) over the 1961–2018 calibration period and the whole study period 1860–2020, as listed in Table (3). For each climate indicator (precipitation), the characteristics of the *reference* (used for the SMHI reference method) and the *calibration* (used in the present EOF method) observation networks are listed in Table (1), and illustrated in Fig. (1).

As a preliminary comment, the EOF (SVD) based ***CI*** estimates are generally very close to the SMHI reference indicator

SMHI-ref. In order to investigate the differences between ***CI*** estimates computed in the present study with SMHI-ref, we introduce $\Delta CI[\mathbf{X_c}]$, the departure from SMHI-ref, defined in Eq. (8).





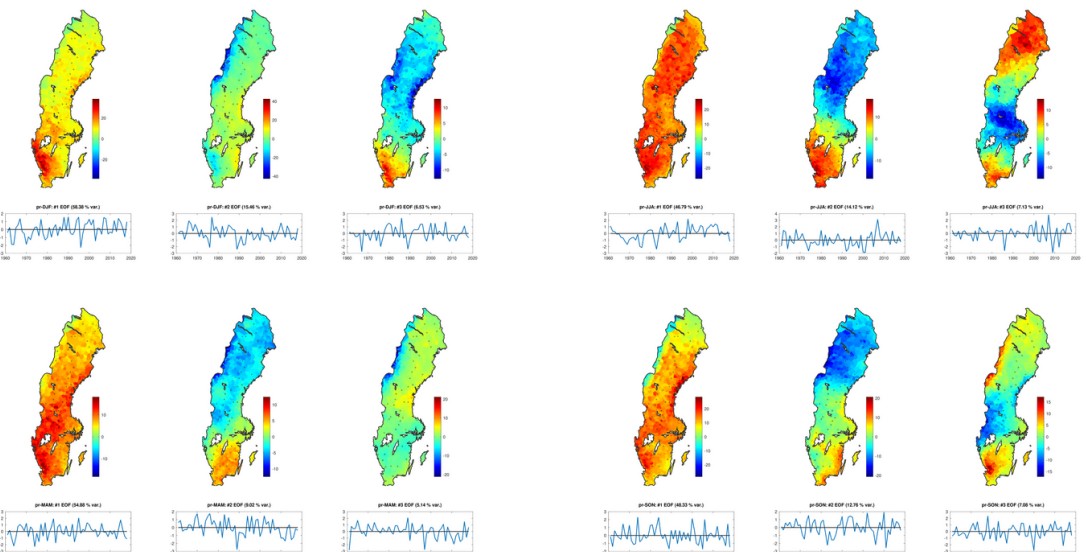

**Figure 5.** Leading three EOF seasonal patterns for MORA *precipitation* over Sweden ($\mathbf{eof}(\#1-3)$, upper row), with their associated time expansion vectors ($\mathbf{a}(\#1-3)$, lower row) for *temperature*. The unit for the product $\mathbf{a}(\#1-3) \cdot \mathbf{eof}^T(\#1-3)$ is in °C. Black crosses indicate the location of MORA stations in the reference network.

The definition of all **CI** methods, including their corresponding legend for figures in this section, are summarised in the upper Table (3). The upper part of the table lists indicators that apply to the 1961–2018 calibration period, while **CI** described in the lower part apply to the entire study period 1860–2020.

$$
\begin{cases}
CI_{ori}^{Ref} & = CI_{ori}[\mathbf{X}^{\star}(\text{Reference network})] \\
\Delta CI[\mathbf{X_c}] & = CI[\mathbf{X_c}] - CI_{\text{SMHI-ref}}^{centered}
\end{cases}
\tag{8}
$$

The calibration period 1961–2018 presents the advantage of having a complete GRIDCLIM dataset, as well as comparing gap-filled $\mathbf{X_c}$ and original $\mathbf{X_c^{\star}}$ MORA datasets. The present section evaluates the performance climate indicator (**CI**) estimates with EOF (respectively SVD) methods. We also assess the representativity of the *reference*, *calibration* MORA station networks for Sweden's climate, compared to a **CI** computed over all grid-points in GRIDCLIM covering Sweden.

Following estimations of **CI** are available over the calibration period:

– $CI_{\text{SMHI-ref}}$: current **CI** definition used by SMHI, calculated as the arithmetic mean of observations for selected MORA observations (i.e. a sub-set of the MORA calibration network). $CI_{\text{SMHI-ref}}$ is not calculated in the present study; only the climate indicator is available, without access to the underlying station data (including station coupling, partial homogenisation and correction). Therefore, the centered climate indicator, in this particular case, is obtained as: $CI_{\text{SMHI-ref}}^{centered} = CI_{\text{SMHI-ref}}^{abs} - \overline{CI_{\text{SMHI-ref}}^{abs}}\Big|_{1961-2018}$



| Label | Mathematical expression and Description | Nb. sta-tions | Date range | Co-lour |
|---|---|---|---|---|
| GridClim-ref | $CI_{ori}[\mathbf{X}_c^{\text{GridClim}}(Reference\,network)]$, arithmetic average for all GRIDCLIM grid-points $\in$ reference network | 87 | 1961-2018 | black |
| GridClim-sub | $CI_{ori}[\mathbf{X}_c^{\text{GridClim}}(Calibration\,network)]$, arithmetic average for all GRIDCLIM grid-points $\in$ calibration network | 1098 | 1961-2018 | yel-low |
| GridClim-all | $CI_{ori}[\mathbf{X}_c^{\text{GridClim}}(Gridpoints\,over\,Sweden)]$, arithmetic average for all GRIDCLIM grid-points $\in$ Sweden | 69,842 | 1961-2018 | green |
| EOF-cal | $CI_{EOF}[\mathbf{X}_c^{MORA}(Calibration\,Network)]$, EOF-based method applied to the gap-filled SMHI-MORA data for the calibration network | 1098 | 1961-2018 | cyan |
| SMHI-ref | $CI_{ori}[\mathbf{X}^{\star}(Reference\,network)] - \overline{CI_{ori}}|_{1961-2018}$, original SMHI indicator, centered to its 1961–2018 mean, for MORA data in the reference network | 87 | 1860-2020 | grey |
| EOF-rec | $CI_{EOF}[\mathbf{X}_c^{\star MORA}(Calibration\,Network)]$, EOF-based method applied to the SMHI-MORA data for the calibration network | 1098 | 1860-2020 | red |

**Table 3.** Description of the labels for Figures with corresponding label and colour. The number of stations for the reference and calibration networks are indicated. The mathematical formalism follows the definitions in the text, where $\mathbf{X}_c^{MORA}$ refer to the centered SMHI-MORA dataset, and $\mathbf{X}_c^{\text{GridClim}}$ refer to the gridded GRIDCLIM dataset.

- $CI_{\text{GridClim-ref}}$: arithmetic average of time-series from the GRIDCLIM dataset for grid-points where *reference* stations are located. $CI_{\text{GridClim-ref}}$ is primarily meant to assess effect of time-space averaging order. Unlike $CI_{\text{SMHI-ref}}$, the spatial average is performed on centered station records $CI_{\text{GridClim-ref}}^{centered} = \dfrac{\sum_{i_{sta}=1}^{n_{sta}} \mathbf{X}^{\text{GridClim}}(ind_{\text{Ref-Network}},:)}{n_{sta}}$

- $CI_{\{\text{Sub,All}\}\text{-GridClim}}$: arithmetic average of time-series from the GRIDCLIM dataset for grid-points where *calibration* stations are located (for $CI_{\text{Sub-GridClim}}$), or all grid-points covering Sweden, as shown in e.g. Fig. (2) (for $CI_{\text{All-GridClim}}$).

- $CI_{\text{EOF-cal}}$: average over the EOF-reconstructed $\widehat{\mathbf{X}_{\text{MORA}}^{\text{EOF}}}$, according to Eq. (6).

The definition of $\boldsymbol{CI}$ over the 1961–2018 calibration period, including their corresponding legends for figures Fig. (8) and following, are summarised in the upper Table (3).

### 3.2.1 Comparing SMHI-ref and GridClim-ref over 1961 – 2018

Before comparing GridClim-sub and GridClim-all with GridClim-ref, let us evaluate GridClim-ref versus SMHI-ref, operationally used by SMHI. Other national weather and climate agencies use gridded dataset to compute a national climate indicator, e.g. for the contiguous US Vose et al. (2014). The method to compute the GridClim-ref $\boldsymbol{CI}$ is similar, but not identical





to the SMHI-ref *CI*. To be consistent with EOF estimates, the GridClim-ref is based on the longest single time-series in the Mora dataset, rather than the coupled 'pseudo-station' used in SMHI-ref (in which nearby stations are 'stiched together' to

obtain longer continuous time-series, as explained in Joelsson et al. (2023). Furthermore, the centered climate indicator for GridClim-ref (and all other *CI* computed in the current study) is computed as the mean of centered station time-series; on the other hand, SMHI-ref is computed as the arithmetic mean of absolute station observations, from which the mean over the 1961–2018 calibration period was subtracted (according to Eq. (8)), since individual station data used for SMHI-ref calculation were no longer available.

The fact that SMHI-ref is computed with absolute values from observation stations makes it more sensitive to inhomogeneities: if, despite the coupling procedure, a given station is interrupted, the computed *CI* will be affected, especially if the multi-annual mean temperature of this station departs significantly from the ensemble mean.

    Over the 1961–2018 calibration period, the GridClim-ref *CI* (in black) can hardly be distinguished from SMHI-ref *CI* (in grey) in Fig. (8): the estimated inter-annual variability (upper panel in Fig. (8)) is virtually identical for GridClim-ref and SMHI-

ref. After applying a 10-year Gaußian running filter (lower panels in Fig. (8)), we find a fair agreement between GridClim-ref and SMHI-ref for precipitation. Fig. (6) enable to look in more details at systematic differences between GridClim-ref and SMHI-ref representing $\Delta CI$, the departure from SMHI-ref ( according to Eq. (8)). *CI* estimates by SMHI-ref and GridClim-ref agree within at least $\Delta CI_{\mathrm{prec}} \in \pm 1$ mm.month$^{-1}$ for precipitation over the 1961–2018 calibration period. This sets a first constraint on the *CI* accuracy, thus a metrics for the performance of EOF-based *CI* estimates.

**3.2.2   How does spatial sampling affect the 1961 – 2018 climate indicator ?**

    Similar to SMHI-ref, we calculated 3 *CI* as arithmetic means: GridClim-ref (discussed in the previous section, with GridClim values at the same locations as the SMHI-ref network), GridClim-sub, with GridClim values at locations corresponding to the Mora calibration network, and finally GridClim-all, using all GridClim grid-points covering the Swedish territory (with the same resolution as Fig. (2)).

GridClim-sub, with 1098 stations for precipitation, shown as the yellow curve labeled GridClim-sub in Fig. (8), is hardly distinguishable from GridClim-ref (i.e. 87 stations for precipitation, as black lines) when considering absolute *CI* values. We can thus conclude that (i) the GridClim dataset is – when averaged over many stations – in good agreement with observed Mora observations, and (ii) estimates for the climate indicator are robust, when reducing the GridClim-sub sampling ensemble from 1098 to 87 carefully chosen members in SMHI-ref, i.e. reducing the sample size by a factor 13. Hence the new method,

based on the larger sample size in GridClim-sub, is likely to be consistent with the SMHI-ref *CI* published by SMHI.

    However, we see a significant change for the arithmetic *CI* in GridClim-all, compared to GridClim-ref (black line) and GridClim-sub (yellow line): the *CI* is computed from all grid-cells covering Sweden (GridClim-all, in green line), stands out with a larger amplitude in inter-annual variability. In other words, annual *CI* values with local minima (maxima) i GridClim-sub appear generally lower (higher) in GridClim-all.

GridClim-all, using all 69 842 GridClim grid-points covering the Swedish territory (c.f. pixel resolution in Fig. (2)) captures more of the inter-annual climate variability than the 87 stations in the SMHI-ref network for precipitation. This is consistent




with the EOF patterns shown on Fig. (3): the 3 dominant modes in Figure Fig. (2) show large variations in remote areas (e.g. Jämtland in central-western Sweden, Norrbotten close to the Finnish border). Such patterns are qualitatively well captured by the calibration network (Fig. (3), upper row): the leading EOF patterns ($\mathbf{eof}^{MORA}(\#1-3)$ are similar to ($\mathbf{eof}^{GridClim}(\#1-3)$, with a strong correlation of their respective time expansion coefficients (lower plots in Fig. (2) and Fig. (3)).

However the difference in amplitude between the GridClim-all and GridClim-ref can be related to the under-sampled remote areas with large variations. Hence an important conclusion: the definition of a 'national' observation-based climate indicator is dependent on the station network on which it relies. In other words, a national network of stations covering most of the territory is not *per se* a guaranty that all regions are equally represented in the 'national' climate indicator, and may differ from a gridded, land-covering approach. Vose et al. (2014) applied a methodology similar to GridClim-all: a spatially interpolated dataset for temperature and precipitation observations for the contiguous US, with elevation-dependent dependence, proved to make a significant difference the previous *CI* version.

The similarity between GridClim-ref, with 87 stations for precipitation, and GridClim-sub with 1098 stations, supports the sub-sampling method used for the *CI* robustness (Equation Eq. (7)). Given that a 10-time sub-sampling of carefully selected stations (i.e. the reference network for GridClim-ref compared to the calibration network for GridClim-sub) yield comparable results, we consider that selecting an ensemble of GridClim-sub subsets, each consisting of a randomly chosen subset with 3-time as many stations as GridClim-ref, is reasonable to evaluate the *CI* robustness.

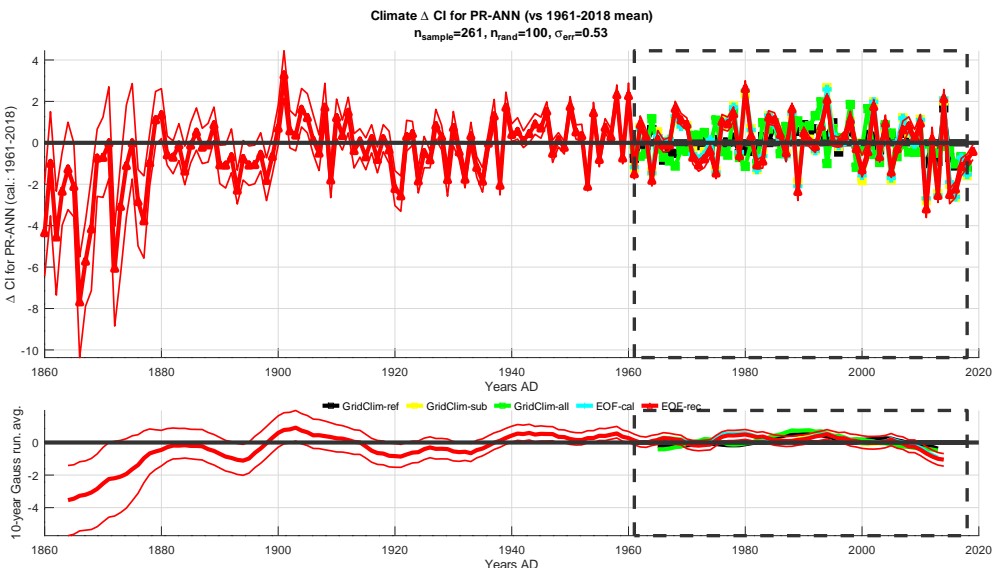

**Figure 6.** $\Delta CI$, i.e. departures from centered original SMHI indicator, for **precipitation** $\Delta CI[\mathbf{X_c}] = CI[\mathbf{X_c}] - CI_{\text{SMHI-ref}}^{centered}$. Labels are identical to Figure Fig. (6) .



For precipitation, the difference between GridClim-ref, GridClim-sub and SMHI-ref are harldly noticeable, while GridClim-all stands out with a generally larger inter-annual variability: local maxima/minima are more pronounced in GridClim-all. Given

the spatially unhomogeneous character of rainfall and larger range of precipitation values (compared to temperature), it is not surprising that GridClim-ref, GridClim-sub and GridClim-all display differences, which are likely related to geographic sampling issues. However, such discrepancies are mostly punctual (for individual years), an do not display a systematic bias in their 10-year averaged values (lower panel). In conclusion, various methods to compute the precipitation *CI* indeed display slight differences, which though are small (both on inter-annual and decadal scale) with respect to the overall *CI* signal.

### 3.2.3    Comparing EOF estimated climate indicators to the SMHI reference indicator for 1860–2020: precipitation

As for temperature, the EOF-based *CI* reconstruction for precipitation is generally in good agreement with SMHI-ref. Fig. (8) illustrates that the precipitation *CI* forthe reference SMHI-ref is in good agreement with EOF-cal (over the 1961–2018 calibration period) and EOF-rec from 1900 onward, both for inter-annual and decadal variability: the EOF-based methods prove to be consistent with the original *CI* for precipitation SMHI-ref. Similar to temperature, the 25%–75% percentile range

associated to EOF-cal is fairly stationary from 1940 onward. Prior to 1940, the robustness for EOF-rec gradually increases, with a sharp increase prior to 1880.

Subtle differences between SMHI-ref and EOF-rec can be identified when investigating $\Delta CI$, i.e. the departure from SMHI-ref (Fig. (6)). Over the period 1880–1960, the $\Delta CI$ for EOF-rec is within the same range as EOF-cal over the 1961–2018 period (namely $\pm 2$ mm.month$^{-1}$). Between 1939–1946, EOF-rec displays a continuous 8-year period during which the *CI*

is consistently higher (by $\Delta CI = 1$ mm.month$^{-1}$) than SMHI-ref. This period occurs shortly after the massive launch of precipitation observations in 1931, where the total number of active stations registered in the MORA database increased from 307 to 600 (Fig. (1)).

During the early part of the record (1860–1901 in Fig. (6)), $\Delta CI$ for EOF-rec displays large departures from SMHI-ref. Year 1901 appears to be a dry year in Sweden; according to SMHI-ref, it stands out as the driest on record ($-20$ mm.month$^{-1}$), while

EOF-rec evaluates it to be $-16$ mm.month$^{-1}$, in line with other extreme dry years (1933, 1947, 1976 or 2018). Interestingly, year 1901 coincides with a noticeable drop of active stations (Fig. (1)), which could explain inhomogeneities in SMHI-ref.

Prior to 1877, the number of observation stations for precipitation is very low: less than 30 in total, 24 of which are included in EOF-rec and 14 in SMHI-ref (Table Table (1)). Accordingly, the EOF-rec $\Delta CI$ displays much larger values ($\pm 6$ mm.month$^{-1}$) than in the later part of the record. Nevertheless, the 10-year Gaußian filter (Figure Fig. (6), lower panel)

shows a consistent increasing trend over the 1860-1880 period. This feature is not surprising: precipitation observations in SMHI-ref were corrected to account for underestimated measurements by ancient rain-gauges (Alexandersson, 1986). The impact of the correction factor is discussed in the next section, especially for winter (DJF) precipitation.

In summary, available instrumental evidence may not be sufficient to determine a robust precipitation *CI* for the 1860–1900 period with the present methodology. 1900–1920 is characterised by dry conditions over Sweden (on average $-7$ mm.month$^{-1}$

drier than the 1961–2018 mean), followed by fairly stationary conditions until 1965 ($-4$ mm.month$^{-1}$ on average). The decade 1968–1977 records several consecutive dry years (e.g. $-12$ mm.month$^{-1}$ in 1976). Precipitation over Sweden has





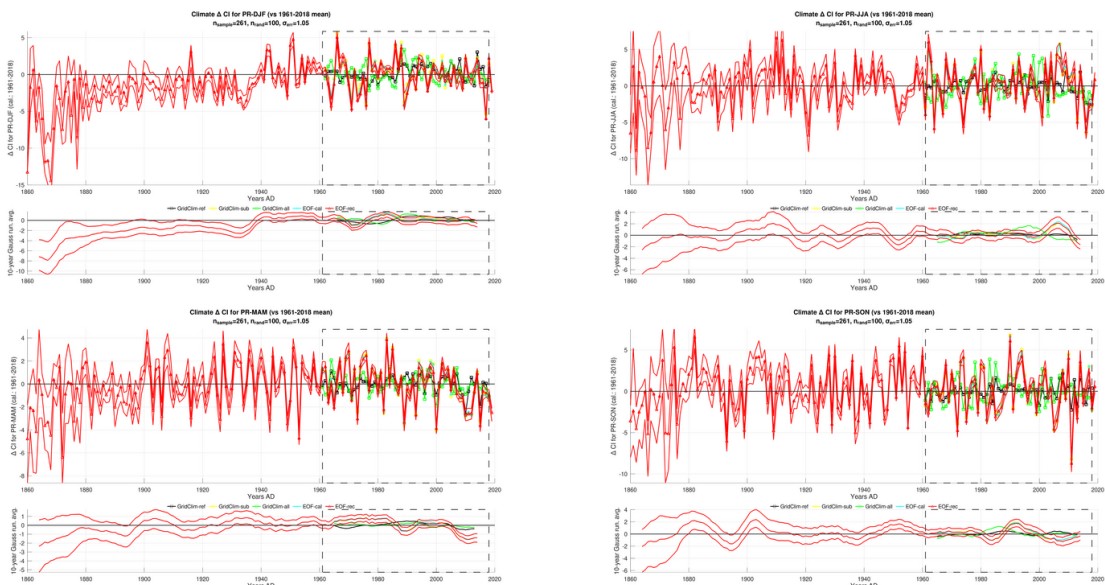

**Figure 7.** $\Delta CI$, i.e. departures from centered original SMHI indicator, for **precipitation** $\Delta CI[\mathbf{X_c}] = CI[\mathbf{X_c}] - CI_{\text{SMHI-ref}}^{centered}$. Labels are identical to Figure Fig. (6) .

experienced a sustained increase over the 1977–2020 period, reaching up to 4 mm.month$^{-1}$ on average for the 2005-2015 decade. Years 2000, 1998 and 2012, with up to 15 mm.month$^{-1}$ wetter than the 1961–2018 mean, stand out as **_CI_** maxima over the instrumental period.

### 3.2.4 Seasonal EOF reconstructions of the climate indicator: precipitation

When analysing seasonal EOF-rec, one needs to keep in mind that the EOF patterns $\overrightarrow{\textbf{eof}^{\text{seas}}}$, defined in Eq. (4), are computed independently over the 1961–2020 period for each seasonal average $\mathbf{X}_c^{\text{seas}}$. Hence the difference for the annual $\Delta CI_{EOF}^{\text{ANN}}$ cannot be obtained as an average of seasonal $\Delta CI_{EOF}^{\text{seas}}$. In other words:

$$\forall iSeas = \text{DJF, MAM, JJA, SON}, \Delta CI_{EOF}^{\text{ANN}} \neq \frac{\sum \Delta CI_{EOF}^{iSeas}}{4} \tag{9}$$

Unlike for temperature, the reconstructed **_CI_** for precipitation (EOF-rec) does display a significant departure from SMHI-ref prior to 1900: EOF-rec is consistently lower than SMHI-ref for annual and seasonal precipitation.

This result is expected: the SMHI-ref record has undergone a correction for early precipitation values (Alexandersson, 1986). The rain gauge designed used by SMHI prior to 1900 is known to underestimate precipitation amounts: the correction factor yields thus higher values then the observed ones.

The data used with the EOF method were extracted from the MORA database that contains actual measurements (i.e. uncorrected). Therefore, EOF-rec cannot be expected to reproduce the corrected precipitations amounts in SMHI-ref.



The present study illustrated the magnitude of the correction: the 10-year smoothed average for the annual $\Delta CI$ is around $-3$ mm.month$^{-1}$, with similar or lesser results for MAM, JJA and SON. The discrepancy is significantly larger for winter precipitation (DJF): it amounts to 8 mm.month$^{-1}$. This result is consistent with the fact that the underestimation of precipitation is largest when precipitation comes as snow, which is common in Swedish winters.

Individual year and seasonal $\Delta CI$ records display large, spurious variations prior to 1900. Analogously to the $\Delta CI$ for temperature, the sparse station coverage prior 1900 causes the EOF method to deviate strongly from SMHI-ref. The inter-annual variability of EOF-rec prior to 1900 is however not deemed to be statistically significant, as illustrated by the large span in 25%–75% percentiles ($\pm 2$ mm.month$^{-1}$).

The 10-year smoothed $\Delta CI_{\text{DJF}}$ for (EOF-rec, Fig. (7)) displays a sharp break after 1934, from a persistent $\Delta CI_{DJF} = -2$ mm.month$^{-1}$ bias to an unbiased estimator afterwards (i.e. $\Delta CI$ 0 mm.month$^{-1}$). The 1934 break is synchronous with a doubling of active precipitation stations, from less than 300 to more than 500 stations (Fig. (1)), while the number of active stations for SMHI-ref remains fairly constant. A likely explanation resides in the representativity of the EOF-rec precipitation network: prior to 1940, the limited observation network, when projected on EOF patterns computed over the 1961–2018 reference period, the DJF $CI$ leads to a underestimation. Such a bias is particularly pronounced for winter (DJF), since the leading mode ($\mathbf{eof}_{\text{DJF}}^{\text{MORA, GridClim}}(\#1)$, shown in Fig. (4) and Fig. (5), are characterised by a regional maximum centered over Sweden's West coast (around Göteborg).

The leading EOF pattern for the annual average and other seasons (MAM, JJA, SON) (Fig. (4)) display a more balanced $\mathbf{eof}(\#1)$. Therefore, the change in the number (and location) of active stations used in EOF-rec does not introduce a persistent bias, as for DJF. The impact of network change is perceivable by smaller variations of $\Delta CI$ around major shifts in the observation network (1934, 1946).

## 4 Discussion

### 4.1 Benefits of the EOF-based climate indicators

Similar to the computation of the temperature $CI$ (Sturm, 2024a), the EOF-based $CI$ estimate EOF-rec for precipitation is in overall agreement with the reference method SMHI-ref (i.e. *arithmetic* mean of a carefully selected subset of MORA observations). This confirms that the 87 stations in the reference network are globally representative of the full (1098) MORA dataset – which is *per se* a novel result, since it had not be proven earlier. Our study also illustrates that the EOF-based precipitation $CI$ EOF-rec differs significantly from the reference method SMHI-ref prior to 1900; the discrepancy is particularly pronounced for winter (DJF) precipitation. This difference is primarily due to a correction factor, meant to compensate the underestimation of precipitation (especially snow) measurements in older rain gauges. The correction is not applied to the EOF-rec $CI$, since it uses actual measurements stored in the MORA database. SMHI is currently leading a study to quantify the precipitation losses in 19$^{\text{th}}$-century gauges used in Sweden (Joelsson, in preparation).

The present study introduces the EOF method as an alternative $CI$ estimate: EOF-rec can be considered as *weighted* average of MORA precipitation observations. The weighting coefficients are determined using independent (i.e. orthogonal) modes





425 of variability in the observations during the calibration period. The weighing coefficients are derived from EOF (empirical orthogonal functions) patterns, noted **eof**.

  We choose a calibration period extending from 1961–2018, which corresponds to the period covered by the gridded climate dataset GRIDCLIM. From a theoretical point of view, using a weighted $CI$ method makes its estimates less sensitive to changes in observation availability over time, i.e. how the spatial density of stations in the observation network evolves over time. Such

430 an automated method also allows to use the full observation database (1098 stations), instead of solely hand-selected stations in the reference network (87 stations).

  Based on the EOF method, we also suggests a metrics to evaluate the range of likely $CI$ values. By adding random noise and sub-sampling in an ensemble computation, we can display the 25% and 75% percentiles in addition to the median value for the complete dataset. Prior to 1900, beyond the impact of the correction factor, the large [25%, 75%] span in $\Delta CI_{EOF-rec}$ (up

435 to $\pm 3$ mm.month$^{-1}$) suggests that validity of a new correction factor (Joelsson, in preparation) might be difficult to assess for inter-annual variability; the evaluation of the 10-year running mean appears to be more robust. However, SMHI-ref and EOF-rec rely on similar MORA observations: it is therefore not trivial to assess the validity of a correction factor. One suggestion is to evaluate trends, inter-annual and decadal variability in river discharge measured in Sweden since at least early 20$^{\text{th}}$ century (Wörman et al., 2010) as observations independent of precipitation.

440 Furthermore, the EOF analysis enables to visualise the dominant modes of variability in the MORA observation dataset, and compare it to the gridded dataset GRIDCLIM. The EOF patterns illustrate that the seasonal variability for precipitation is significantly different from annual means.

### 4.2 Implication for decadal climate variability over Sweden

Fig. (6) and Fig. (7)) illustrates a steady increase in annual precipitation over Sweden since 1940. Precipitation increase is

445 particularly pronounced during winter (DJF) and spring (MAM), while autumn (SON) displays no distinctive trend over the 1961–2018 calibration period. Unlike other seasons, summer (JJA) precipitation exhibits a $-10$ mm.month$^{-1}$ dip between 1960–1970, followed by a steady increase ($+20$ mm.month$^{-1}$) until 2010. SMHI-ref and EOF-rec yield similar results, with the exception of winter (DJF) during 1900–1930 where EOF-rec is lower by $-2$ mm.month$^{-1}$. The latter results in an increase of winter precipitation of $+20$ mm.month$^{-1}$ for EOF-rec versus $+15$ mm.month$^{-1}$ for SMHI-ref. It is however unclear how

450 much of these differences are related to data correction applied in SMHI-ref.

### 4.3 Comparison to climate indicators in other countries

Vose et al. (2014) presents the climate division dataset for the conterminous US for temperature and precipitation since 1895. Station data is interpolated using thin-plate smoothing spline method, accounting for invariant predictors such as elevation. A national indicator can be derived from area-weighted average of "division units" constituting the grid.

455 The method for deriving a area-weighted climate indicator for the contiguous is analog to the GridClim-all $CI$ presented in this study, i.e. the mean of all grid-points in the GRIDCLIM dataset covering Sweden.



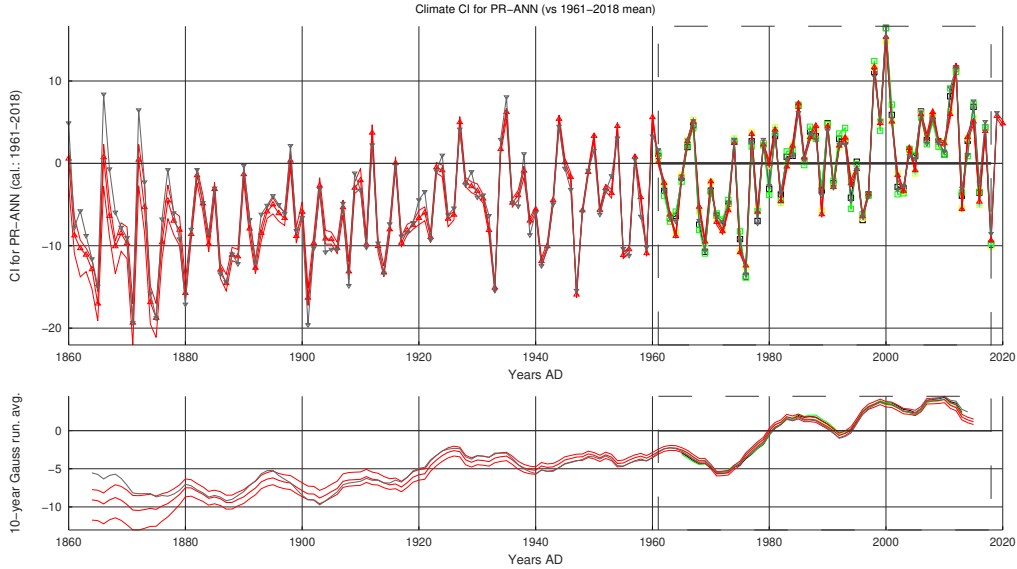

**Figure 8.** Estimates of the annual climate indicator **CI** for **precipitation** with various methods. The **CI** is represented as thick lines, whereas the 25% and 75% percentiles of the robustness ensemble $CI_{ens}$ are shown as thin lines of the same colour. The labels for various **CI** are indicated in Table Table (3) .

The difference between the new and old versions for the monthly precipitation **CI** by Vose et al. (2014) indicates that version 2 is wetter than version 1 by $5\,mm$ water equivalent. Prior to 1931, the difference increases to $15\,mm$ water equivalent, reflecting a change of the computational method used in version 1.

Konstali and Sorteberg (2022) analyses a network of 55 homogeneity-tested observation stations since 1900, compare to a denser network of 199 stations since 1960. Climate indicators are first calculated for 8 "precipitation regions", before being aggregated to a national indicator. The distinction in regions allows to correct precipitation according to the pseudo-adiabatic ascent model, which is a particular requirement for the steep relief across Norway. The methodology applied in Norway is thus more advanced than SMHI's, enabling to assess changes of the climate indicators across different regions. Among other
methods, the SVD approach (as described in the next section) can provide consistent regional climate indicators over time, despite changes in the observation network.

Meteo-Swiss Begert et al. (2005) provides a set of 12 homogenised monthly precipitation series since 1864. Climate indicators are available as time-series for each reference observation station, or as interpolated maps over the Swiss territory (since 1871). The Swiss meteorological service adopts a hybrid approach (maps and individual observation time-series), which differs
from SMHI's aggregated national climate indicator.

Kaspar et al. (2017) presents a review of climate observation in Germany since 1881. Available observations are interpolated on a 1 km x 1 km grid, which allows for altitude-dependent correction of the considered climate variable.



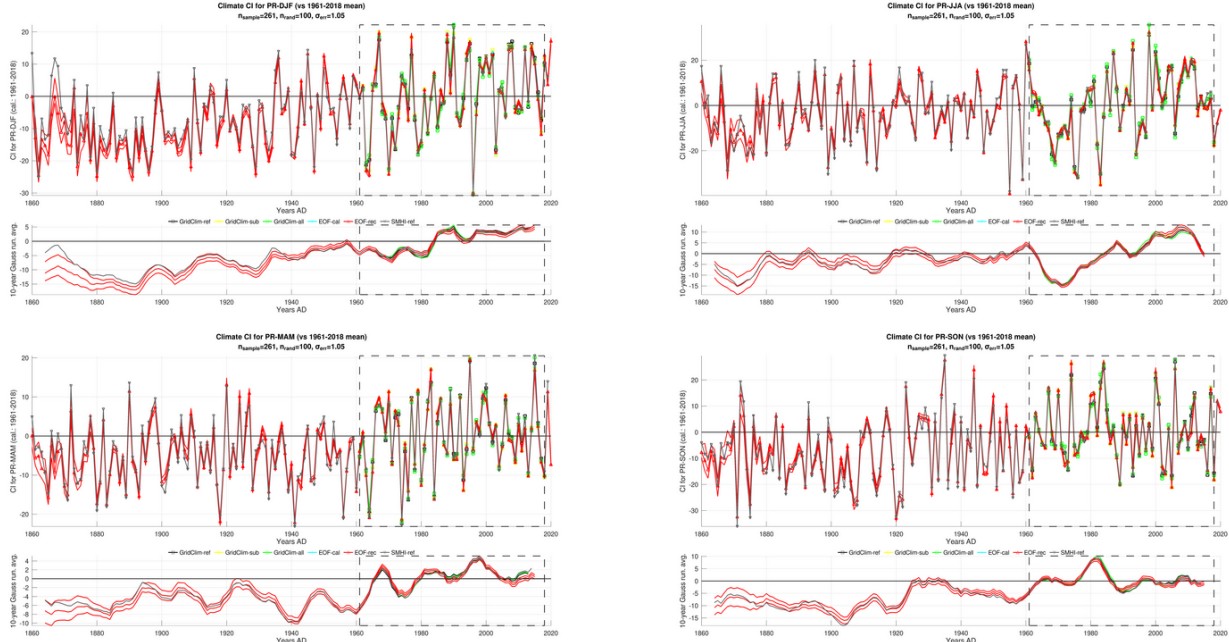

**Figure 9.** Reconstructed seasonal absolute *CI* for **precipitation** and the centered original SMHI indicator $\Delta CI[\mathbf{X_c}] = CI[\mathbf{X_c}] - CI_{\text{SMHI-ref}}^{centered}$. Labels are identical to Figure Fig. (8) .

In the United Kingdom (Office, 2024), climate indicators are available for England, Wales, Scotland and Northern Ireland, as well as the entire UK since 1881, based on the HadUK-grid. In France (Meteo-France, 2024), monthly indicators are available for each department since 1852, as well as long homogenised time-series for selected observation stations.

This brief overview reveals that most national weather agencies adopt a hybrid approach to produce climate indicators. Long, homogenised time-series provide a long-term base-line for climate change in time, which is complemented by gridded datasets to represent its spatial variability. The present study shows the relevance of the EOF method to better capture the spatial of climate variability, as a first step towards delivering region-specific climate indicators. The homogenisation of climatological observations is an ongoing task at SMHI (Joelsson et al., 2022, 2023).

Räisänen and Alexandersson (2003) conclude that the 1991–2000 decade was warm and wet in Sweden, which has a 93% probability to be related to anthropogenic climate change. These results are consistent with Fig. (8), which also show that the warm and wet trend has continued over the 2000–2020 period. It is therefore relevant to investigate if the leading EOF modes $\mathbf{eof}^{MORA}$ are affected by increasing greenhouse gas levels. This example further underlines how pertinent a national climate indicator for precipitation can be, in order to quantify and illustrate the effect of anthropogenic climate change in Sweden.



## 4.4 Persistence of leading EOF modes

Kjellström et al. (2022) presents large-scale atmospheric clustering, and how they affect Scandinavian precipitation between two climate normals (1961–1990 and 1991–2020). The frequency of circulation types display significant changes between 1961–1990 and 1991-2020; the change in occurrence frequency differs also for individual months. Kjellström et al. (2022) reports strong inter-annual correlations between the North Atlantic Oscillation (NAO) index and temperature $\textit{CI}$, in particular for winter (DJF) with an $R^2 = 60\%$; the $\textit{CI}$ for summer (JJA) temperature however is much smaller $R^2 = 5\%$. For precipitation, the winter (DJF) $\textit{CI}$ has the strongest correlation ($R^2 = 20\%$). It is worth noticing that the correlation coefficients between the NAO-index and the national precipitation $\textit{CI}$ , for each season, are very close between the 1961–1990 and 1991-2020 period.

As for temperature (Sturm, 2024a), results by Kjellström et al. (2022) raise the question to which extent the leading EOF modes $\mathbf{eof}(\#1-10)$ vary over various time-slices. In particular, the EOF-based reconstruction $\mathsf{EOF\text{-}rec}$ relies on the assumption that EOF patterns derived during the 1961–2018 period correctly reproduce the internal modes of variance for precipitation over the 1880–1960 period. Further research is thus needed to evaluate if the leading EOF modes $\mathbf{eof}(\#1-10)$ for precipitation vary significantly over 30-year periods. In that case, the EOF method could be adapted by identifying a predictor representing circulation types (e.g. based on mean sea-level pressure reconstructions since 1850 (Allan and Ansell, 2006; Ansell et al., 2006; Gallego et al., 2005)). A particular set of EOF patterns $\mathbf{eof}(\#1-10)$ can be assigned to dominant clusters of the predictor, which could be used for $\textit{CI}$ reconstruction $\mathsf{EOF\text{-}rec}$ for earlier periods.

The North Atlantic Oscillation (NAO) is commonly accepted as the dominant circulation type affecting climate variability in Sweden, in particular during winter. The fact that correlation $R^2$ between the NAO-index and precipitation time-series is similar between 1961–1990 and 1991-2020 in Kjellström et al. (2022) indicate that, despite a change in the occurrence of circulation type, the relation between NAO and precipitation $\textit{CI}$ is persistent. In other words, while the circulation types change, the internal variability (i.e. the leading EOF patterns) remains mostly unchanged.

The long-term evolution of precipitation regimes over Scandinavia (Chen et al., 2021) shows an increasing trend of the 1880-2020 period, whose magnitude is unprecedented since 1550. Observation data presented in this study can be used to assess the correlation of circulation indices with precipitation time-series over Sweden, for annual and seasonal averages, over 30-year time-slices between 1860 and 2020. Several reconstructions of NAO since 1850 are available (Cropper et al., 2014, 2015; Comas-Bru and Hernández, 2018; Hanna et al., 2022; Jacobeit et al., 2001b; Slonosky et al., 2000; Zveryaev, 2006), or the last millennium (Gouirand et al., 2007; Jones et al., 2001). Comas-Bru and Hernández (2018) also provides estimates of the East Atlantic (EA) and Scandinavian (SCA) circulation patterns since 1851 to present. We however would advise that observations in the MORA database be processed at a monthly resolution (instead of annually resolved ANN, DJF, MAM, JJA, SON averages) for such an analysis. The present EOF method can be adapted, possibly improved, by computing leading EOF modes $\mathbf{eof}$ from monthly results instead of annual/seasonal averages, which might better capture the short-term variability of NAO.

Jacobeit et al. (2001a, 2003) present zonal circulation indices (NAO: North Atlantic Oscillation, CEZ: Central European Zonal Index) for Europe over the 1780–1995 period. The NAO index shows a period until the 1850s with accumulating negative



anomalies in winter whereas, in summer, positive are prevailing during 1820–1910. This is in contrast with recent evolution
with positive anomalies prevailing during winter (1920–1970) and negative ones during summer (1970–1995). Numerous
studies have assessed the impact of synoptic circulation on (mostly NAO) on precipitation patterns across Scandinavia since
1850 (Hänsel, 2020; Ansell et al., 2006; Gallego et al., 2005; Hanna et al., 2022; Alvarez-Castro et al., 2018; Fleig et al., 2015;
Jones and Mann, 2004; Philipp et al., 2007; Slonosky et al., 2000; Kyselý, 2007, 2008). While it is beyond the scope of the
present study to assess the dependence of observed precipitation to synoptic circulation patterns, it would be interesting to
investigate whether the leading EOF modes $\mathbf{eof}^{MORA}$ differ significantly for e.g. positive/negative phase composites of the
NAO.

Such an analysis would however require to analyse the precipitation observations at monthly resolution, instead of annually
resolved (ANN, DJF, MAM, JJA, SON) as in the present study: Massei et al. (2007), based on a wavelet analysis of the daily
NAO index, demonstrate that a large portion of the power-spectrum of NAO variability is found at sub-annual frequencies.

## 5   Conclusions

The present study introduces a new method to aggregate observation time-series into a single climate indicator, which is
evaluated for precipitation over Sweden. A prerequisite for the new method is to emulate the climate indicator operationally
used by the Swedish Meteorology and Hydrology Institute (SMHI), based on an arithmetic mean of all station time-series.

### 5.1   Added value of station coupling and homogenisation

The present study focuses on the individual station measurements in the MORA database maintained by SMHI. On one hand,
it has the advantage of considering all available observations; on the other hand, it comprises only few station with long time-
series. Climatologists (including the authors of the SMHI-ref indicator) have used the 'coupling' technique to stitch together
similar station records into a long, single 'pseudo-station'; however, such a process has long been a time-intensive, person-
operated effort; it has thus been restricted to a limited number of carefully chosen stations (e.g. the SMHI reference network).
A recent development by Joelsson et al. (2022) enables an automatic, objective coupling routine. The present methodology can
be improved by using the coupling routine prior to applying the EOF method.

The quality control applied to all observations stored in the MORA database aims at insuring that the instrumental value
is correctly digitised; however it does not account for biases related to changes in station location, instruments, surroundings
etc. An *a posteriori* correction can be performed using homogenisation, such as BaRT/Homer or Climatol (used at SMHI).
Homogenisation is particularly recommend when working with coupled records, since the 'stitching' is likely to introduce
variations that are not related to climate, hence considered inhomogeneities.

Further developments of the present method will investigate the impact of coupling and homogenisation on instrumental ob-
servations on the estimation of the climate indicators. An automated method for station coupling (Joelsson et al. (2022, 2023))
enables an optimal compromise between the maximum number of different time-series and the longest times-series. The cou-
pling procedure will increase the number of time-series retained in the calibration dataset EOFcal that extend to the early



parts of the record. in other words, the light grey line in Fig. (1) is likely to be closer to the total number of active stations (represented as staples). Furthermore, the number of missing values during the 1961–2018 calibration period is likely to be reduced, hence the gap-filling procedure (prior to the EOF analysis) is less likely to introduce numerical artefacts in the indicator

reconstruction.

However, possible inhomogeneities (introduced by the coupling and/or other events, e.g. station relocation) will remain in the calibration dataset EOFcal. These can be removed with the homogenisation procedure (e.g. using Climatol (Guijarro, 2024) and/or Bart/HOMER toolboxes (Joelsson et al., 2023)). This additional step will illustrate to which extent the national climate indicator is sensitive to (presumably randomly distributed) inhomogeneities.

Finally, the homogenisation toolboxes Climatol and Bart/HOMER include a gap-filling feature (based on similarities with neighbouring stations), which potentially can deliver complete time-series over the entire experiment period 1860–2020. It is however questionable to which extend the gap-filling algorithm delivers physically significant results over such long periods, when the station availability drops to low numbers (e.g. in the early part of the record). The present method, in particular the ensemble computation with randomly distributed noise, would be a suitable tool to evaluate the robustness of the indicator

estimation depending on the pre-processing steps (station coupling and homogenisation (Joelsson et al., 2022)), the choice of the calibration period and the climate variable (temperature, precipitation and potentially other observational datasets).

### 5.2 Further development of the SVD approach

As illustrated earlier, the EOF and SVD methods are largely equivalent in the present objective: defining a *CI* primarily based on the MORA station network. However, the SVD method has the potential to be developed further. Eq. (A3) only makes use of

$\mathbf{SVD}^{MORA}$ (defined in Eq. (A1)) to estimate $\widehat{\mathbf{X}}_c^{SVD}$, the MORA dataset over 1860–2020. The SVD method allows to use the other term in Eq. (A1): the spatial pattern $\mathbf{SVD}^{\text{GridClim}}$ and associated time expansion coefficients $\mathbf{A}^{\text{GridClim}}$. By construction (Björnsson and Venegas, 1997), the SVD time expansion coefficients $\mathbf{A}^{MORA}$ and $\mathbf{A}^{\text{GridClim}}$ are related to each other, with eigenvalue matrix $\Lambda^{SVD}$ defined in Eq. (A1).

$$\Lambda^{SVD} = \mathbf{A}^{MORA} \cdot \left(\mathbf{A}^{\text{GridClim}}\right)^T$$
$$\mathbf{A}^{\text{GridClim}} = \left(\left(\mathbf{A}^{MORA}\right)^{-1} \cdot \Lambda^{SVD}\right)^T$$

where the notation $\left(\mathbf{A}^{MORA}\right)^{-1}$ refers to a pseudo-inverse, since $\mathbf{A}^{MORA}$ is not a square matrix. Based on this result, the SVD method makes it possible to estimate a reconstructed gridded dataset $\widehat{\mathbf{G}_c}^{\text{GridClim}}$:

$$\widehat{\mathbf{G}}_c^{SVD} = \widehat{\mathbf{A}}^{\text{GridClim}} \cdot \left(\mathbf{SVD}^{\text{GridClim}}\right)^{-1}$$
$$\Leftrightarrow \widehat{\mathbf{G}}_c^{SVD} = \left(\left(\widehat{\mathbf{A}}^{MORA}\right)^{-1} \cdot \Lambda^{SVD}\right)^T \cdot \left(\mathbf{SVD}^{\text{GridClim}}\right)^{-1}$$
$$\Leftrightarrow \widehat{\mathbf{G}}_c^{SVD} = \left(\left(\mathbf{X}_c^\star \cdot \mathbf{SVD}^{MORA}\right)^{-1} \cdot \Lambda^{SVD}\right)^T \cdot \left(\mathbf{SVD}^{\text{GridClim}}\right)^{-1}$$

(10)



New climate indicators based on $\mathbf{G}_c^{SVD}$ can potentially be used to emulate the GridClim-all over the entire 1860–2020 period, provided that the projection of MORA $\mathbf{X}^\star$ on its SVD patterns is sufficient to capture the (dominant) climate variability

over the entire Swedish territory. If that is the case, it would be possible to define regional climate indicators based on $\mathbf{G}_c^{SVD}$, even in regions where no direct measurements are available. This assumption might be hard to fulfil in early parts of the records, where only few stations are available (cf. Fig. (1)), and would require additional research to assess the significance of the method. The present method could extended by using regional climate simulations over the 1860–2020 period in substitution for the single GRIDCLIM dataset used in this study; the principle of applying SVD to combine (even sparse) observations with

a gridded dataset remains valid. This is a promising way to characterise Sweden's pre-industrial climate, to the best precision the available observation evidence allows.

This example illustrates that relatively simple linear algebra methods, such as EOF and SVD, have the potential to deliver valuable results for the analysis of climate observations. The analysis can be further refined using more advanced method, such as Principal Oscillation Patterns (POP), as presented in Storch et al. (1995).

*Code and data availability.* The code and processed data used in this study are freely available on Zenodo (Sturm, 2024b).

The computations for the present study are performed using the open-source software OCTAVE, which is mostly compatible with MATLAB. The code is run with OCTAVE version 7.3.0 on a Linux computer. It further requires following OCTAVE packages:

  – *io*, version 2.6.4

  – *netcdf*, version 1.0.16

– *stat*, version 1.4.3

  – *mapping*, version 1.4.1

The OCTAVE code is available freely available at. Since the final processed data used in this study is available (cf. below), the user may be able to run the code without the full list of required OCTAVE packages.

The data, processed as required for this study, is freely available for download. The archive format is OCTAVE's binary format, which

can be loaded into an OCTAVE session using the *load* command. The processed data files, including the result of the EOF computation, is organised in separate files for each annual and seasonal means (annual – ANN, and seasonal: winter – DJF, spring – MAM, summer – JJA, autumn – SON).

## Appendix A:  Supplementary material





## A1  Number of seasonally active stations

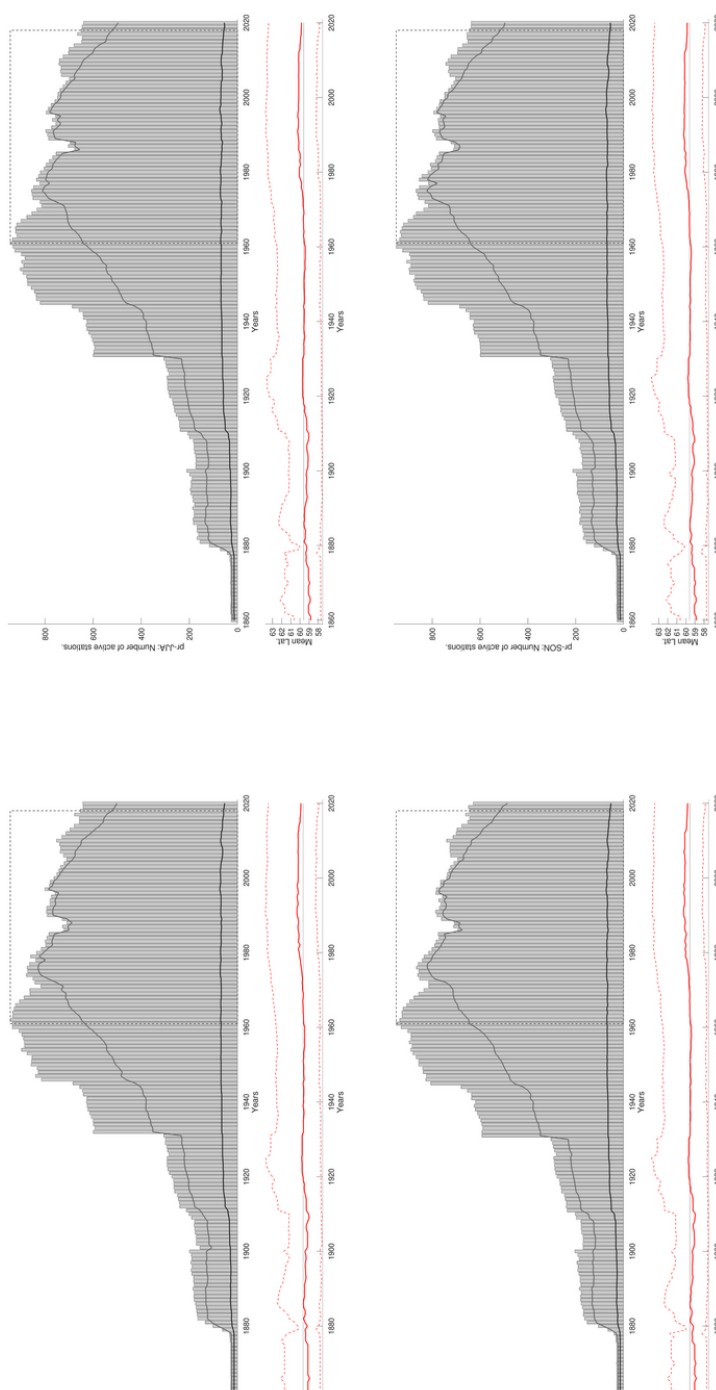

**Figure A1.** Upper plot: Number of active precipitation stations in MORA over time (as bars), for each season (cf. Fig. (1)). The dark grey line represents the number of active in the original reference station network; they light grey line represents the number of stations for the calibration network (i.e. individual stations being active at least 15 years during the calibration period 1961–2018, as highlighted by the dashed box). Lower plot: Median latitude for active stations in the calibration dataset over time (incl. the [25%−75%] bounds). The median latitude is used as a proxy for the distribution of the observation network, in particular its coverage of Sweden's Norther regions.



## A2 SVD decomposition of the coupled MORA and GRIDCLIM datasets

In section 2.2.2, the EOF decomposition was described for a single independent dataset (MORA or GRIDCLIM). The EOF method can be further developed using the Singular Value Decomposition (SVD) method described hereafter. In the present study, the leading EOF patterns $\mathbf{eof}(\#1-3)$ were, in all but one case, virtually identical to their SVD counterparts $\mathbf{svd}(\#1-3)$ for both the MORA calibration network and the complete GRIDCLIM dataset.

The "Singular Value Decomposition" (SVD) is a further development of the EOF method (Eq. (3)), where the leading modes no longer express the dominant variability in a single dataset, but the shared variability modes that two datasets have in common. The SVD requires that both datasets have the same number of time-steps (i.e. the same number of columns), but may have different spatial dimensions (i.e. number of rows).

Being able to jointly assess the common variability, over the same period, of two datasets with different spatial extent is a particular benefit of the SVD method. In the present case, the (climate) variability of a given MORA station is no longer solely compared to its corresponding grid-cell in the GRIDCLIM dataset (e.g. as was done in the gap-filling procedure); the SVD enables to assess how the 1098 MORA time-series for observed precipitation relate linearly to the $69\,842$ individual time-series from GRIDCLIM grid-cells covering the Swedish territory. Since regional-scale climate variability is (presumably) well captured in the GRIDCLIM re-analysis product, the SVD method thus enables to isolate corresponding trends in the MORA dataset – regardless of how well the one-to-one correspondence between the observed and simulated signal matches for any single location.

Covariance matrix: $\mathbf{R} = \mathbf{X}_c^{MORA} \cdot \left(\mathbf{X}_c^{\text{GridClim}}\right)^T$

Singular Value Decomposition: $\mathbf{R} = \mathbf{SVD}^{MORA} \cdot \Lambda^{SVD} \cdot \left(\mathbf{SVD}^{\text{GridClim}}\right)^T$

Time Expansion Coefficients:

$$\begin{cases} \mathbf{A}^{MORA} = & \mathbf{X}_c^{MORA} \cdot \mathbf{SVD}^{MORA} \\ \mathbf{A}^{\text{GridClim}} = & \mathbf{X}_c^{\text{GridClim}} \cdot \mathbf{SVD}^{\text{GridClim}} \end{cases}$$

(A1)

Analogously to Eq. (3), the portion of the explained (common) variance can be retrieved from eigenvalues in diagonal matrix $\Lambda^{SVD}$. Similar to Eq. (4), the original $\mathbf{X}_c^{MORA}$ and $\mathbf{X}_c^{\text{GridClim}}$ matrices can be identically reconstructed from the results of the SVD:

$$\begin{cases} \mathbf{X}_c^{MORA} = \mathbf{A}^{MORA} \cdot \left(\mathbf{SVD}^{MORA}\right)^T \\ \mathbf{X}_c^{\text{GridClim}} = \mathbf{A}^{\text{GridClim}} \cdot \left(\mathbf{SVD}^{\text{GridClim}}\right)^T \end{cases}$$

(A2)

The first three leading modes of the SVD for the MORA calibration subset are presented in Fig. (A2): the top row shows the spatial SVD patterns $\left(\mathbf{svd}^{MORA}\right)^{[1:3]}$, the bottom row the corresponding time expansion coefficients $\left(\overrightarrow{\mathbf{a}^{MORA}}\right)^{[1:3]}$. The portion of the variance explained by each mode is obtained from eigenvalues in matrix $\mathbf{\Lambda^{SVD}}$, and displayed in the legend.





Analogously to Eq. (5), the original dataset with missing values $\mathbf{X}_c^\star$ can estimated using SVD patterns as $\widehat{\mathbf{X}}_c^{SVD}$.

$$\widehat{\mathbf{X}}_c^{SVD} = \widehat{\mathbf{A}}^{MORA} \cdot \left(\mathbf{SVD}^{MORA}\right)^{-1} \tag{A3}$$

The climate indicator $\boldsymbol{CI}$ can thus be computed according to Eq. (6). Since $\mathbf{eof}^{MORA}$ and $\mathbf{svd}^{MORA}$ are virtually identical, the reconstructed EOF-rec and SVD-rec cannot be distinguished from each other. Hence SVD-rec is not shown on Fig. (6) and related figures.



## A3 Leading SVD patterns for SVD-rec

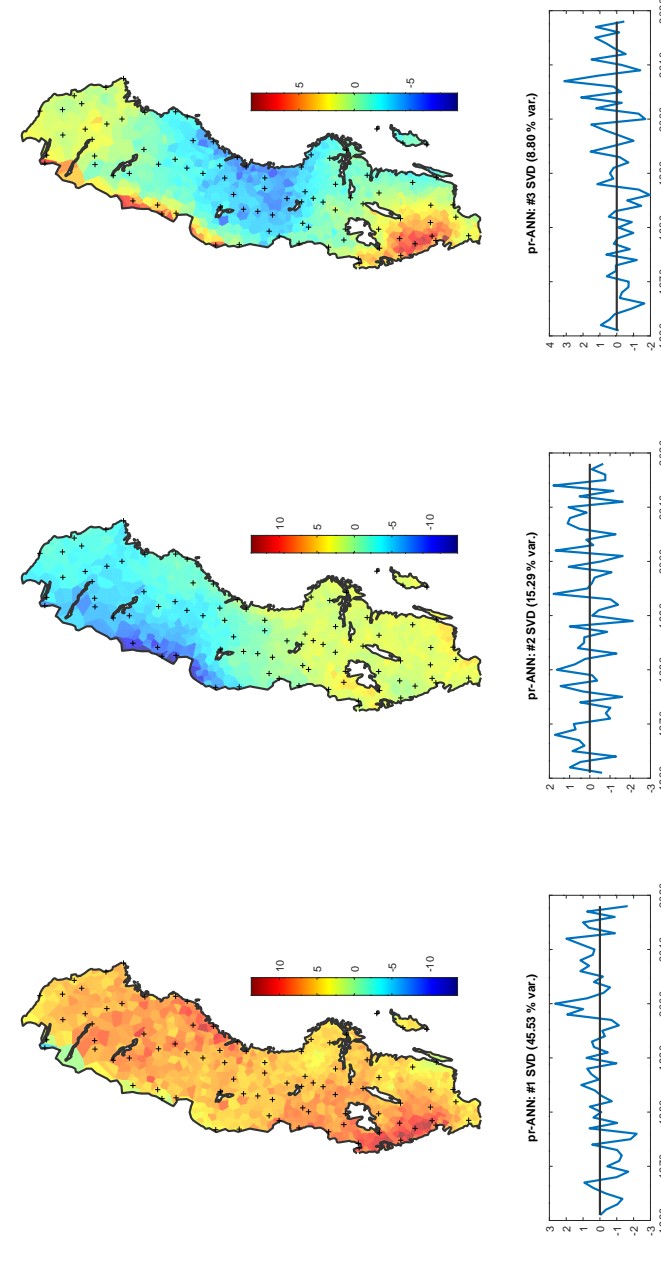

**Figure A2.** Leading three EOF patterns for MORA (Calibration network) over Sweden ($\mathbf{eof}^{MORA}(\#1-3)$), with their associated time expansion vectors ($\mathbf{a}_{EOF}^{MORA}(\#1-3)$) for precipitation.





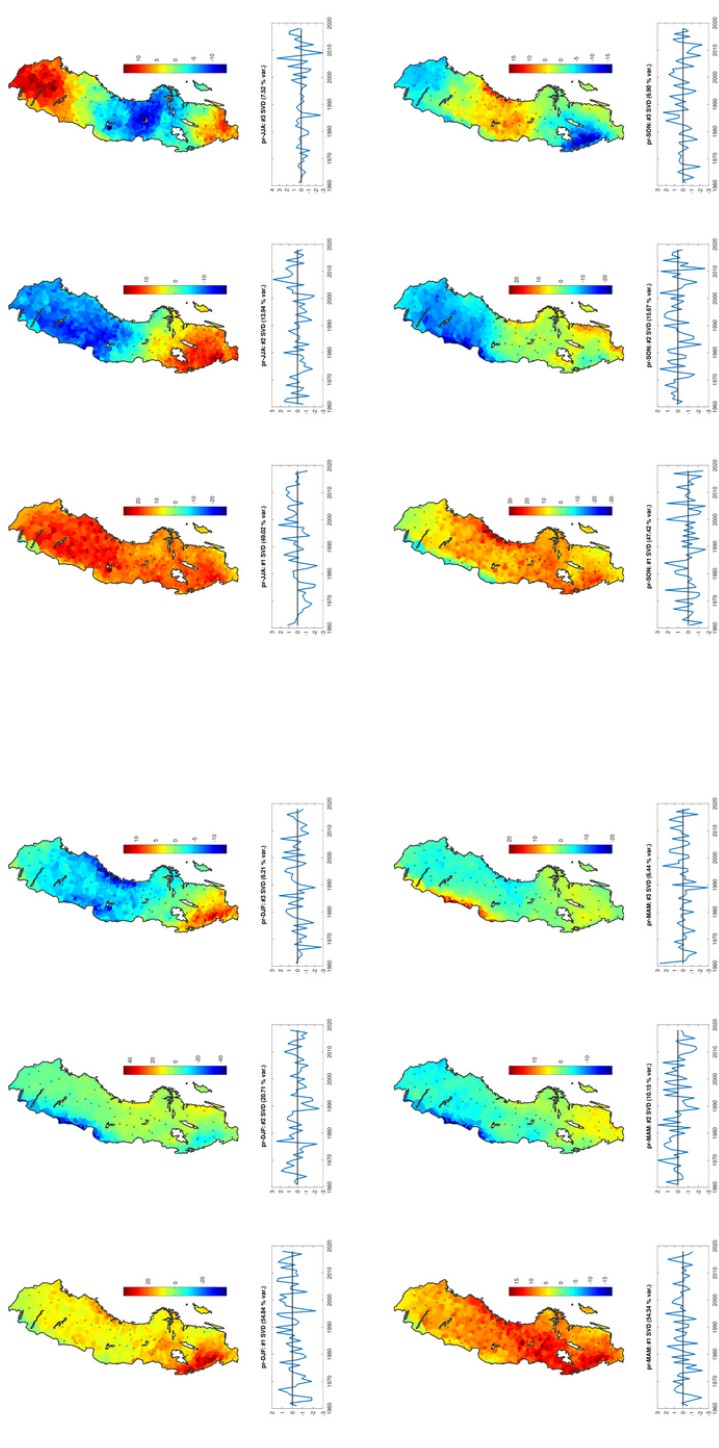

**Figure A3.** Analogue to Figure Fig. (A2) , for seasonal SVD patterns of *precipitation* instead annual. Seasons are defined as DJF (winter, upper left), JJA (summer, upper right), MAM (spring, lower left) and SON (autumn, lower right).



*Author contributions.* The current study was designed, performed and written by the first (and only) author.

*Competing interests.* The author declares that they have no conflict of interest.

*Acknowledgements.* The manuscript has greatly benefited from comments and suggestions by Peter Berg, Erik Engström, Semjon Schimanke and colleagues. Furthermore, no evaluation of historic measurements over Sweden would be possible without the dedication of Lennart Wern, Sverker Hellström and colleagues to digitise, quality-check and organise observations in the MORA database.



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
