# Peer review of "Spatially aggregated climate indicators over Sweden (1860–2020), part 2: Precipitation"

_EGUsphere, 2024_

## Referee Comment (RC2)

**Review:  Spatially aggregated climate indicators over Sweden (1860–2020), part 2: Precipitation**

The paper presents an interesting method for presenting past precipitation in Sweden, but the paper has not been carefully explained. It's hard to follow and seems to lack a logical 'red thread'. I found it unstructured and the explanations are difficult, as much text assumes prior knowledge about the data and analysis. Indeed, the manuscript seems to be unfinished, and not really ready for submission. It should also to a greater extent acknowledge relevant past work (e.g. use Scholar Google).

While it's good to show mathematical expressions, it may be a bit overwhelming in this paper as they are not explained and hard to follow with many different terms/variables. Also perhaps move more of it to an appendix with clearer explanations?  No need to repeat reference to SVD every time EOF is mentioned.

The paper would be more interesting if it not only looks at total precipitation, but also on the wet-day frequency ($f_w$) and the wet-day mean precipitation ($\mu$). The total precipitation is the product of the two and the number of days: $X = n\, f_w\, \mu$ and these two parameters are useful for many purposes (e.g. [1]).

I will advise the author to do a final language, text and editorial check of the document to weed out some minor quirks, typos and linguistic imperfections. Also, the paper could refer to more relevant work on EOFs/PCA (e.g. use Scholar Google) since these methods  are not so new (e.g. [2]). Also, the figure quality wasn't the best - blurred images when zooming in on the principal components.

Again, I found this manuscript a bit annoying as the text was difficult to follow and I find it a bit tedious after a while, especially with repetitive high level of (mathematical) details that with benefit can be presented in an appendix. It's also a long manuscript that becomes a struggle to read, as it throws a lot of facts at me but doesn't really say why it's relevant. I get the impression that the text has not been properly reviewed before submission. Parts of the text in the discussion could perhaps be moved to the introduction section, but it should also be explained why it's relevant for this analysis - rather than presenting them as facts. Please state why these earlier works are referred to here.
* * *
[1] Benestad et al., "Testing a Simple Formula for Calculating Approximate Intensity-Duration-Frequency Curves."

[2] Benestad et al., "Geographical Distribution of Thermometers Gives the Appearance of Lower Historical Global Warming."

I also miss some evaluation of the methods, e..g. carried out with 'pseudo-observations' (e.g. take climate model data and extract series that correspond to the locations of the observational network and remove parts to simulate gaps of missing data - but for which we have complete knowledge about their evolution - see e.g. [2]).

I recommend **major revisions** (e.g.checking the language, improving the figures, restructure and shortening the main text - also back up several statements with proper references and citations).

**More detailed comments**

L30. Please avoid abbreviating terms such as climate indicator (CI) since it makes the text harder to read. Common abbreviations are OK but uncommon ones demand that the reader keep them in their mind while reading and trying to understand the text.

L35: Does the de-correlation distance vary with time scales, e.g. daily, monthly, seasonal and annual means?

L45: Reference to EOFs? E.g. [3]

L54: Method for doing what?

L113: Explain SVD (and give references, e.g.[4])

L120: It may be useful to repeat the details of the method in an appendix, rather than referring to a companion paper (it's a bit annoying).

Section 2.2.2 can be moved to an abstract.

L181: There is no need to assume a normal distribution when the data is present and can be tested for it.

L182: Is the daily observational error as large as 10 mm?

L205: The second and third modes do not necessarily have to be bi-modal and tri-modal, but they have to be orthogonal.  Were the EOFs calculated on anomalies or the total amounts, BTW?

L219: Unclear sentence: "*hence, even for precipitation, the portion of variance is deemed sufficient to reconstruct…*"
* * *
[3] Lorenz, "Empirical Orthogonal Functions and Statistical Weather Prediction."
[4] Strang, *Linear Algebra and Its Application*; Wilks, *Statistical Methods in the Atmospheric Sciences*.

L224: Perhaps "tend to experience less precipitation **variations** than the country's average"?

Fig.2: State whether these maps are for the annual precipitation in the figure captions. Also it would be good to present the eigenvalues to see if these modes are well separated[5] (the same goes for the subsequent figures).

Fig.3: Perhaps a clarification (this is probably already explained, but hidden in a lot of details) Is MORA a gridded product, or were the station series gridded somehow? A distinction between principal component analysis (PCA) and EOFs is that the former is applied to an irregular spatial distribution of series while the latter is applied to data on a regular grid with grid-box area weighting[6]. It's possible to grid the spatial patterns from PCA to get a product similar to EOFs, albeit with slightly different weighting of the various locations (more weight to regions with more densely populated stations) - is this what Fig.3 shows? Or is it merely the projection of EOFs of GRIDCLIM over the calibration period onto the observational station series? Also, has cross-validation been used for evaluation of individual station series (e.g. leaving one station out in calibration and then using it for comparison)?

L242: Emphasise that the modes in the previous section were annual as opposed to seasonal.

Fig.4: Label the panels with the seasons.

L258: Sentence incomplete…

Fig.6: Not easy to see all the curves.

L416: The term 'globally' may be a bit confusing since this study is about Sweden.

L432: Adding random noise - how can it provide information about ensemble spread? What's the nature of the said random noise? White noise or AR(1)? How sensitive are the estimates to the type of noise?

L444-450: Is it possible to say if the trends are due to changing number of wet days or changed intensity?

L458: What is meant by "*is wetter than version 1 by 5 mm water equivalent*"?

L463: Please provide references to the methodology applied in Norway.

L467: Meteo-Swiss Begert et al.?

L481: What does this really mean: "*has a 93% probability to be related to anthropogenic climate change*"?

L497: Such tests can be conducted on climate model results (CMIP or CORDEX).
* * *
[5] North, Bell, and Cahalan, "Sampling Errors in the Estimation of Empirical Orthogonal Functions."
[6] Benestad et al., "On Using Principal Components to Represent Stations in Empirical-Statistical Downscaling."

L490: Why discuss the temperature? Isn't that the 'companion paper'?

L508: "*whose magnitude is unprecedented since 1550*" ?!? Where did this year come from? Such statements need to be backed up with references.

L513: Who is 'We'? (this paper has a single author).

L536: I'm confused: "*On one hand, it has the advantage of considering all available observations; on the other hand, it comprises only few station with long timeseries.*"

The Conclusions section is long and rambling - it resembles a discussion section. It should be concise and sum up the main findings presented in this paper. Also, using a mathematical expression with unorthodox notations (for SVD, it's common to use 'X = U Λ V$^{T}$' - also SVD are used in two different contexts, which makes it more confusing: as a way to estimate EOFs and a method to relate two different datasets together) doesn't help - move it to an appendix.

L589: How can POPs improve this?

**References**

Benestad, R. E., H.B. Erlandsen, A. Mezghani, and K. M. Parding. "Geographical Distribution of Thermometers Gives the Appearance of Lower Historical Global Warming." *Geophysical Research Letters*, July 6, 2019. https://doi.org/10.1029/2019GL083474.

Benestad, Rasmus E., Deliang Chen, Abdelkader Mezghani, Lijun Fan, and Kajsa Parding. "On Using Principal Components to Represent Stations in Empirical-Statistical Downscaling." *Tellus A* 67, no. 0 (October 21, 2015). https://doi.org/10.3402/tellusa.v67.28326.

Lorenz, E. N. "Empirical Orthogonal Functions and Statistical Weather Prediction." Sci. rep. Cambridge, Massachusetts: Department of Meteorology, MIT, USA, 1956. https://eapsweb.mit.edu/sites/default/files/Empirical_Orthogonal_Functions_1956.pdf.

North, G. R., T. L. Bell, and R. F. Cahalan. "Sampling Errors in the Estimation of Empirical Orthogonal Functions." *Monthly Weather Review* 110 (1982): 699–706.

Strang, G. *Linear Algebra and Its Application*. San Diego, California, USA: Harcourt Brace & Company, 1988.

Wilks, D. S. *Statistical Methods in the Atmospheric Sciences*. Orlando, Florida, USA: Academic Press, 1995.